# Inhibition of a transcriptional repressor rescues hearing in a splicing factor–deficient mouse

Yoko Nakano[1,2], Susan Wiechert[1,2], Bernd Fritzsch[3], Botond Bánfi[1,2,4,5]

In mechanosensory hair cells (HCs) of the ear, the transcriptional repressor REST is continuously inactivated by alternative splicing of its pre-mRNA. This mechanism of REST inactivation is crucial for hearing in humans and mice. *Rest* is one of many pre-mRNAs whose alternative splicing is regulated by the splicing factor SRRM4; *Srrm4* loss-of-function mutation in mice (*Srrm4$^{bv/bv}$*) causes deafness, balance defects, and degeneration of all HC types other than the outer HCs (OHCs). The specific splicing alterations that drive HC degeneration in *Srrm4$^{bv/bv}$* mice are unknown, and the mechanism underlying SRRM4-independent survival of OHCs is undefined. Here, we show that transgenic expression of a dominant-negative REST fragment in *Srrm4$^{bv/bv}$* mice is sufficient for long-term rescue of hearing, balancing, HCs, alternative splicing of *Rest*, and expression of REST target genes including the *Srrm4* paralog *Srrm3*. We also show that in HCs, SRRM3 regulates many of the same exons as SRRM4; OHCs are unique among HCs in that they transiently down-regulate *Rest* transcription as they mature to express *Srrm3* independently of SRRM4; and simultaneous SRRM4–SRRM3 deficiency causes complete HC loss by preventing inactivation of REST in all HCs. Thus, our data reveal that REST inactivation is the primary and essential role of SRRM4 in the ear, and that OHCs differ from other HCs in the SRRM4-independent expression of the functionally SRRM4-like splicing factor SRRM3.

## Introduction

REST is a transcriptional repressor of hundreds of genes that are expressed selectively in mature neurons and hair cells (HCs). These genes are derepressed in neurons and HCs during development because REST activity is down-regulated dramatically in both cell types (Chong et al, 1995; Schoenherr & Anderson, 1995; Nakano et al, 2018). However, the mechanisms underlying this down-regulation differ in the two cell types. In neurons REST is down-regulated predominantly by transcriptional silencing, whereas in HCs the down-regulation is caused by splicing of an alternative exon of *Rest* (i.e., exon 4) into the mature mRNA (Fig 1A) (Ballas et al, 2005; Nakano et al, 2018). This alternative splicing event shifts the translational reading frame in *Rest* and inactivates the encoded protein by truncating it upstream of a repressor domain and several zinc finger domains that are critical for gene silencing activity (Magin et al, 2002). Genetic evidence from both humans and mice indicates that this exon 4–dependent inactivation of REST is crucial for hearing. Specifically, in humans, the deafness-causing *DFNA27* mutation creates a reading frame-preserving version of exon 4 of *REST* by relocating the splice acceptor site of this exon (Nakano et al, 2018). In mice, genetic deletion of exon 4 (*Rest$^{+/\Delta Ex4}$*) via homologous recombination results in deafness, balance defects, and perinatal degeneration of all HCs (Nakano et al, 2018).

One factor that has been shown to regulate the splicing of exon 4 of *Rest* (as well as ~55 exons of other pre-mRNAs) in the vestibular HCs (VHCs) of the ear is SRRM4 (Fig 1A) (Nakano et al, 2012). This splicing factor is a Ser/Arg-rich protein expressed selectively in HCs and neurons (Calarco et al, 2009; Nakano et al, 2012). Loss of SRRM4 function due to a deletion mutation in the Bronx waltzer mouse line (*Srrm4$^{bv/bv}$*) is associated with balance defects, deafness, and perinatal degeneration of nearly all VHCs as well as ~75% of inner HCs (IHCs) (Nakano et al, 2012). In contrast, the outer HCs (OHCs) remain morphologically intact in these mice. Thus, although *Srrm4* is expressed in all HCs, the effect of the *Srrm4$^{bv/bv}$* mutation on cell survival differs in OHCs versus IHCs and VHCs. The specific splicing alterations that cause HC degeneration in *Srrm4$^{bv/bv}$* mice were not identified before this study, and the extent to which the alteration in *Rest* splicing drives the pathogenesis of deafness and balance defects in *Srrm4$^{bv/bv}$* mice has not been determined.

We previously reported that neurons do not require SRRM4 for the inclusion of exon 4 of *Rest* in the mature mRNA (Nakano et al, 2019). In these cells, SRRM4 and the related protein SRRM3 act redundantly to regulate the alternative splicing of *Rest* until *Rest* is transcriptionally silenced at around postnatal day (P) 16. SRRM3 is encoded by a REST target gene, and it is expressed selectively in neurons and HCs (Scheffer et al, 2015; Nakano et al, 2019). In the

[1]Department of Anatomy and Cell Biology, Carver College of Medicine, University of Iowa, Iowa City, IA, USA [2]Inflammation Program, Carver College of Medicine, University of Iowa, Iowa City, IA, USA [3]Department of Biology, College of Liberal Arts and Sciences, University of Iowa, Iowa City, IA, USA [4]Department of Otolaryngology–Head and Neck Surgery, Carver College of Medicine, University of Iowa, Iowa City, IA, USA [5]Department of Internal Medicine, Carver College of Medicine, University of Iowa, Iowa City, IA, USA

Correspondence: botond-banfi@uiowa.edu

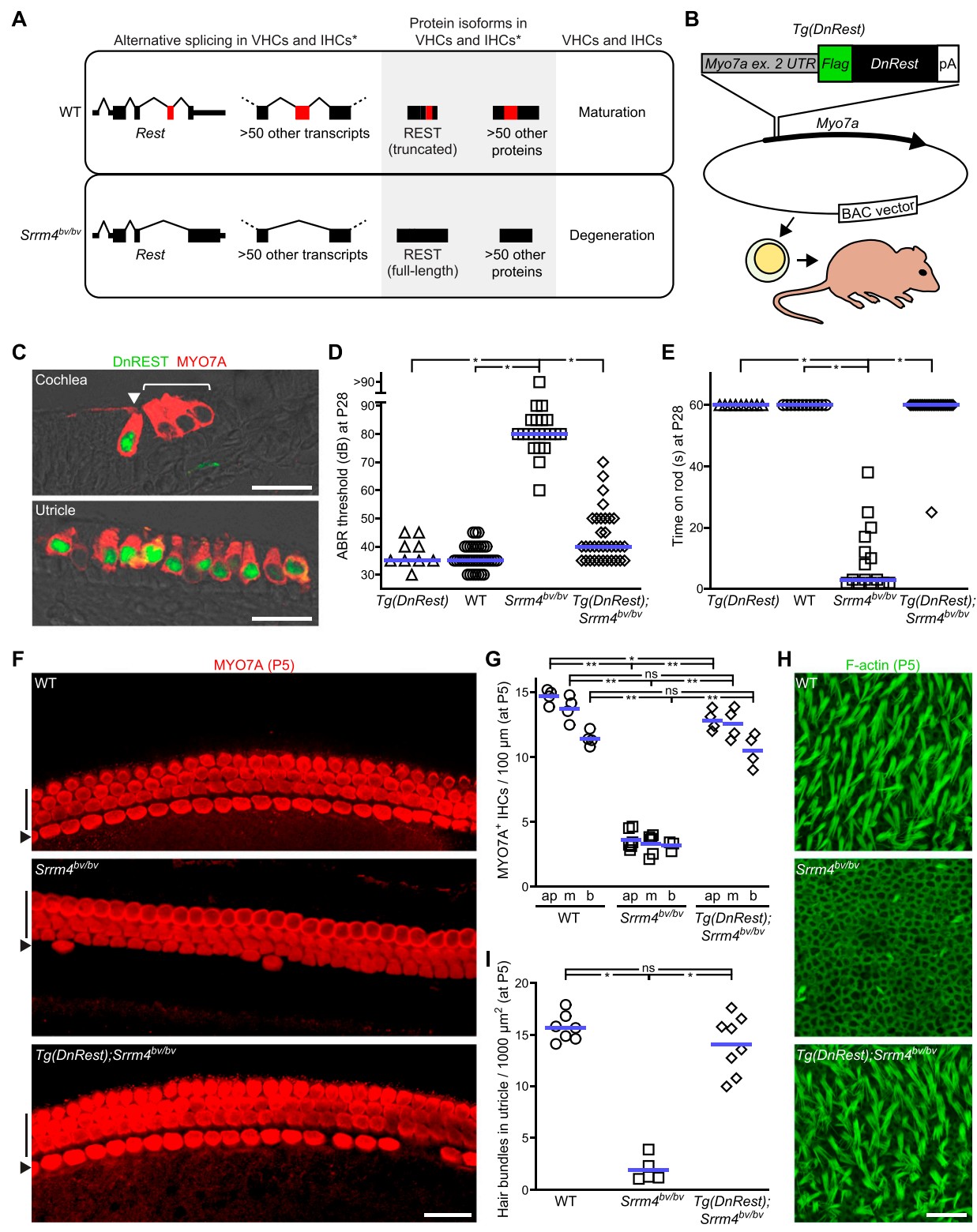

**Figure 1.  *Tg*(*DnRest*) rescues inner ear function, inner hair cells (IHCs), and vestibular hair cells in *Srrm4*[bv/bv] mice.**
**(A)** Schematic of differences in pre-mRNA splicing, protein isoform expression, and hair cell development in wild-type (WT) versus *Srrm4*[bv/bv] mice. Asterisks indicate that the mRNA and protein defects in molecularly characterized vestibular hair cells and molecularly uncharacterized IHCs of *Srrm4*[bv/bv] mice are predicted (but not demonstrated) to be similar. Left column: Diagrams of joining (tented lines) of constitutive exons (black rectangles) and alternative exons (red rectangles) in *Rest* and over 50 other pre-mRNAs. Thin rectangles represent UTRs. Middle: Diagrams of protein isoforms encoded by the transcripts depicted in the left column. Regions encoded by constitutive exons (black) and alternative exons (red) are indicated. Right: Cellular outcomes in WT versus *Srrm4*[bv/bv] mice. **(B)** Schematic of production of *Tg*(*DnRest*)

brain, SRRM3 is required for Purkinje cell survival (Nakano et al, 2019); in the ear, its role has not been evaluated.

Here, we set out to test whether the deafness of $Srrm4^{bv/bv}$ mice is related to the lack of REST inactivation in these animals. We found that transgenic expression of a dominant-negative fragment of REST (DnREST) in HC of $Srrm4^{bv/bv}$ mice rescued hearing, balancing, IHCs, and VHCs, and that it also restored the expression of REST target genes and the alternative splicing of many SRRM4-regulated exons, including exon 4 of $Rest$. Given that $Srrm3$ was one of the DnREST-up-regulated genes in IHCs and VHCs, we examined the possibility of reciprocal regulation between SRRM3 and REST. Our analyses revealed that SRRM3 supported the alternative splicing of $Rest$ in the HCs of $DnRest$ transgenic ($Tg[DnRest]$) $Srrm4^{bv/bv}$ mice and that $Srrm3$ expression was regulated differently in OHCs versus other HCs. Specifically, whereas in VHCs and IHCs the SRRM4-dependent inactivation of REST was critical for the expression of $Srrm3$, in OHCs a transient down-regulation of $Rest$ transcription at birth led to up-regulation of $Srrm3$ independent of SRRM4. Thus, our data show that inactivation of REST is the primary and essential role of SRRM4 in IHCs and VHCs and that OHCs differ from IHCs and VHCs with respect to the configuration of regulatory interactions among SRRM4, REST, and SRRM3.

# Results

### $Tg$($DnRest$) rescues inner ear function and HCs in $Srrm4^{bv/bv}$ mice

To test whether forced suppression of REST activity improves HC survival in $Srrm4^{bv/bv}$ mice, we produced a $Tg$($DnRest$) construct (Figs 1B and S1A). This construct was engineered to contain a DnREST-encoding sequence and a polyadenylation site downstream of the HC-specific promoter of the $Myo7a$ gene, within a bacterial artificial chromosome (BAC). In addition, a Flag tag encoding sequence was inserted in-frame with the DnREST-encoding region to facilitate detection of the DnREST protein (Fig 1B). The Flag tag did not prevent DnREST-dependent inhibition of REST in HEK293 cells (Fig S1B). Transgenic founder mice were generated via pronuclear injection of the $Tg$($DnRest$) construct, and newborn mice from $Tg$($DnRest$) breeding pairs were tested for the expression of Flag-tagged DnREST using an immunofluorescence-based strategy. This analysis demonstrated that DnREST was expressed in the IHCs and VHCs, but not the OHCs, of $Tg$($DnRest$) mice (Figs 1C and S1C), consistent with the patterns of high expression of other $Myo7a$ promoter-driven transgenes (Boëda et al, 2001; Li et al, 2020). Morphological analysis of HCs in the cochlea and utricle of $Tg$($DnRest$) mice did not reveal pathological alterations (Fig S1D and E), suggesting that $Tg$($DnRest$) is not damaging to HCs.

The $Tg$($DnRest$) mouse line was outcrossed to the $Srrm4^{bv/bv}$ mouse line, and the effects on hearing in the second filial (F2) generation were evaluated by measuring auditory brainstem responses (ABRs) to broadband sounds on P28 and P120. These ABR measurements confirmed that $Srrm4^{bv/bv}$ mice were hearing impaired and revealed that, in contrast, the hearing of most $Tg$($DnRest$);$Srrm4^{bv/bv}$ mice was similar to that of wild-type (WT) and $Tg$($DnRest$) littermates (Figs 1D and S1F). To assess the ability of $Tg$($DnRest$);$Srrm4^{bv/bv}$ mice to balance, we measured the time they remained on a fixed horizontal rod. The performance of $Tg$($DnRest$); $Srrm4^{bv/bv}$ mice was similar to that of WT and $Tg$($DnRest$) animals, which remained on the rod for the duration of the assay (60 s). In contrast, most $Srrm4^{bv/bv}$ mice did not remain on the rod for more than 20 s (Figs 1E and S1G). Thus, $Tg$($DnRest$) rescues hearing and the ability to balance on a fixed rod in the $Srrm4^{bv/bv}$ mouse line.

We next analyzed the survival of IHCs in $Tg$($DnRest$);$Srrm4^{bv/bv}$ mice and non-transgenic littermates at P5, the time beyond which the HC loss of $Srrm4^{bv/bv}$ mice does not progress (Whitlon et al, 1996). Organs of Corti were dissected from these mice, and whole-mount preparations of the isolated organs were immunostained with an antibody against the HC protein MYO7A (Figs 1F and S1H). This analysis revealed that more IHCs survived in $Tg$($DnRest$); $Srrm4^{bv/bv}$ mice than in $Srrm4^{bv/bv}$ littermates (Fig 1G). To retest the extent of IHC survival in $Tg$($DnRest$);$Srrm4^{bv/bv}$ mice independently of MYO7A expression, we visualized the stereocilia bundles of HCs and other F-actin–rich structures in the cochlea at P28 using fluorescently labeled phalloidin (Fig S1I). The F-actin staining confirmed that more IHCs in $Tg$($DnRest$);$Srrm4^{bv/bv}$ mice than in $Srrm4^{bv/bv}$ littermates retained stereocilia bundles. We next evaluated the number of stereocilia bundles in the utricles of transgenic and non-transgenic $Srrm4^{bv/bv}$ mice at P5 by F-actin staining, and found more stereocilia bundles in the $Tg$($DnRest$);$Srrm4^{bv/bv}$ versus $Srrm4^{bv/bv}$ utricles (Figs 1H and I and S1J). These data indicate that $Tg$($DnRest$) prevents the degeneration of most IHCs and VHCs in $Srrm4^{bv/bv}$ mice.

mice. Top: DNA elements inserted into mouse $Myo7a$ in a bacterial artificial chromosome. Rectangles indicate UTR of exon (ex.) 2 of $Myo7a$ (gray), Flag tag–encoding region (green), DnREST-encoding region (black), and polyadenylation signal (white). Middle: position of modified exon 2 (vertical lines) in $Myo7a$ (arrow) in the bacterial artificial chromosome (ellipse). Bottom: depiction of injection of transgenic construct into zygotes and transplantation of zygotes into pseudopregnant mice. **(C)** Immunofluorescence detection of Flag-tagged DnREST (green) and MYO7A (red) in cochlear (top) and utricular (bottom) sections from $Tg$($DnRest$) mice at P1 (n = 4 mice). Arrowhead, IHC; bracket, outer hair cells. Scale bars, 20 $\mu$m. **(D)** Thresholds of broadband click-evoked auditory brainstem responses in WT, $Srrm4^{bv/bv}$, and $Tg$($DnRest$);$Srrm4^{bv/bv}$ mice at P28. Each symbol represents the value for a single mouse; blue lines indicate medians (Kruskal–Wallis test $P < 0.05$, Dunn's posttest *$P < 0.0001$, control group: $Srrm4^{bv/bv}$). **(E)** Time spent on a horizontal rod before falling for WT, $Srrm4^{bv/bv}$, and $Tg$($DnRest$);$Srrm4^{bv/bv}$ mice at P28. Maximal duration of the assay was 60 s. Each symbol represents the value for a single mouse; blue lines indicate medians (Kruskal–Wallis test $P < 0.0001$, Dunn's posttest *$P < 0.0001$, control group: $Srrm4^{bv/bv}$). **(F)** MYO7A immunofluorescence in whole-mount preparations of middle-turn region of organ of Corti from WT, $Srrm4^{bv/bv}$, and $Tg$($DnRest$);$Srrm4^{bv/bv}$ mice at P5 (n is shown in Fig 1G). The anti-MYO7A antibody labels IHCs (arrowheads) and outer hair cells (vertical lines). Scale bar, 20 $\mu$m. **(G)** Numbers of MYO7A immunolabeled cells in IHC row of organ of Corti from P5 mice of the indicated genotypes. Data are shown for the apical (ap), middle (m), and basal (b) turn regions of organ of Corti. Each symbol represents value for a single mouse; blue lines indicate means (two-way ANOVA $P < 0.0001$ for genotype factor, Tukey's posttest *$P = 0.041$, **$P < 0.0001$; ns, not significant). **(H)** F-actin–stained utricular macula from WT, $Srrm4^{bv/bv}$, and $Tg$($DnRest$);$Srrm4^{bv/bv}$ mice at P5 (n is shown in Fig 1I). Scale bar, 20 $\mu$m. **(I)** Numbers of cells with stereocilia (hair) bundles in utricles of mice of the indicated genotypes at P5. Each symbol represents value for a single mouse; blue lines indicate means (Welch's ANOVA $P < 0.0001$, Dunnett's T3 posttest *$P < 0.0001$; ns, not significant).

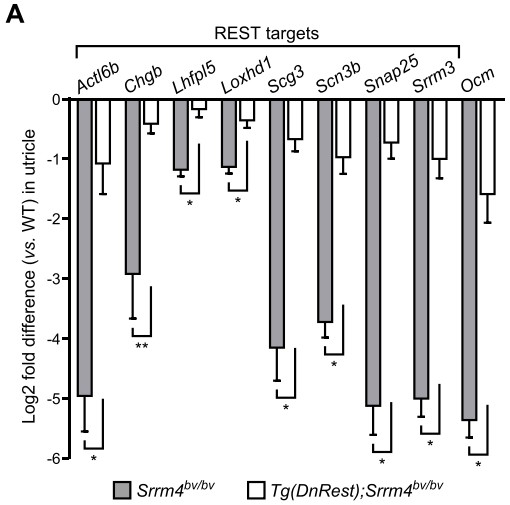

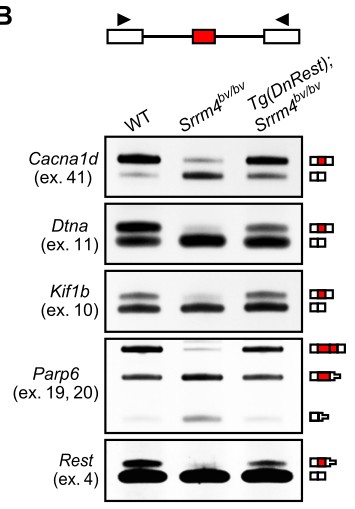

**Figure 2. _Tg(DnRest)_ rescues the expression and alternative splicing of several transcripts in the _Srrm4^{bv/bv}_ utricle.**
**(A)** qRT-PCR data for utricular expression of the indicated genes in _Srrm4^{bv/bv}_ and _Tg(DnRest)_; _Srrm4^{bv/bv}_ mice versus WT mice, at E16.5. Bracket indicates REST target genes. Values are mean ± SEM (n = 4–5 mice per genotype, Welch's unpaired _t_ test, false discovery rate–adjusted *$P < 0.05$, **$P < 0.01$).
**(B)** RT-PCR data for alternative splicing of the indicated exons (ex.) in the utricles of E16.5 WT, _Srrm4^{bv/bv}_, and _Tg(DnRest)_;_Srrm4^{bv/bv}_ mice (n = 3 mice per genotype). The primers (arrowheads) were designed from constitutive exons (white boxes) flanking the tested alternative exons (red boxes). Exons included in specific products are depicted next to gel images; reduction in box height indicates that a stop codon is present.
Source data are available for this figure.

### _Tg(DnRest)_ rescues gene expression and alternative splicing in the utricles of _Srrm4^{bv/bv}_ mice

We tested the effects of _Tg(DnRest)_ on utricular gene expression in _Srrm4^{bv/bv}_ mice by qRT-PCR. Oligonucleotide primers were designed to anneal with the transcripts of nine genes that are known to be expressed at abnormally low levels in the balance organs of _Srrm4^{bv/bv}_ mice (Fig 2A) (Nakano et al, 2012). Of the nine selected genes, eight had been identified as direct target genes of REST based on chromatin immunoprecipitation sequencing (ChIP-seq) (ENCODE Project Consortium, 2012). In contrast, the ninth (_Ocm_) does not contain either a ChIP-seq–identified binding site for REST or a sequence with significant similarity to the REST-binding DNA motif and is thus deemed not to be a target gene of REST (Bruce et al, 2004; Otto et al, 2007). Our qRT-PCR analysis revealed that the utricular expression of all nine genes was higher in _Tg(DnRest)_;_Srrm4^{bv/bv}_ and WT mice than in _Srrm4^{bv/bv}_ mice at embryonic day (E) 16.5 (Fig 2A), that is, before the onset of VHC loss in the _Srrm4^{bv/bv}_ mouse (Fig S2A) (Cheong & Steel, 2002). Thus, _Tg(DnRest)_ rescues the utricular expression of several REST target genes in _Srrm4^{bv/bv}_ mice. In addition, the _Tg(DnRest)_-dependent up-regulation of _Ocm_ indicates that the effect of _Tg(DnRest)_ is not limited to direct target genes of REST.

We next used RT-PCR to test the effect of _Tg(DnRest)_ on the alternative splicing of six SRRM4-regulated exons in _Srrm4^{bv/bv}_ mice. Oligonucleotide primers were designed to anneal with constitutive exons upstream and downstream of the tested alternative exons (Fig 2B). These primers were applied to RNA samples isolated from the utricles of WT, _Srrm4^{bv/bv}_, and _Tg(DnRest)_;_Srrm4^{bv/bv}_ mice at E16.5. Assessment of the RT-PCR products revealed that the inclusion of all six alternative exons in their respective mRNAs was higher in the utricles of _Tg(DnRest)_; _Srrm4^{bv/bv}_ mice and WT mice than in those of _Srrm4^{bv/bv}_ mice (Figs 2B and S2B). Thus, _Tg(DnRest)_ rescues the alternative splicing of several SRRM4-regulated exons in the utricle of _Srrm4^{bv/bv}_ mice. This is surprising because DnREST and REST are not known to regulate pre-mRNA splicing directly. Given that DnREST derepresses the splicing factor–encoding gene _Srrm3_ in the utricle of _Srrm4^{bv/bv}_

mice (Fig 2A), the ability of DnREST to rescue alternative splicing could be due to SRRM3.

### OHCs differ from IHCs and VHCs with respect to SRRM4-independent expression of _Srrm3_

We previously showed that in cortical neurons SRRM4 and SRRM3 act redundantly in regulating the alternative splicing of _Rest_ (Nakano et al, 2019). However, in the utricle, SRRM4 is critical for the alternative splicing of _Rest_ (Fig 2B), likely because utricular expression of _Srrm3_ is minimal in the absence of SRRM4 (Fig 2A). To test whether the cochlear expression of _Srrm3_ also depends on SRRM4, we used duplex in situ hybridization. Probes complementary to _Srrm3_ and the HC-specific mRNA _Myo6_ were hybridized to cochlear sections prepared from WT, _Srrm4^{bv/bv}_, and _Tg(DnRest)_; _Srrm4^{bv/bv}_ mice at P0. Analysis of the _Myo6_ hybridizations demonstrated that in all genotypes the cochlear apex contained IHCs, but in _Srrm4^{bv/bv}_ mice the basal turn of the cochlea lacked nearly all IHCs (Figs 3A and S3A). These data are consistent with the base-to-apex progression of IHC loss in _Srrm4^{bv/bv}_ mice near the time of birth (Whitlon et al, 1996). Analysis of the _Srrm3_ hybridizations at the cochlear apex revealed that the expression of _Srrm3_ was abnormally low in the IHCs (but not OHCs) of _Srrm4^{bv/bv}_ mice. In _Tg(DnRest)_;_Srrm4^{bv/bv}_ mice, the _Srrm3_ hybridization signal was strong in both the IHCs and OHCs (Figs 3A and S3A). These data indicate that SRRM4-dependent inactivation of REST is necessary for the expression of _Srrm3_ in IHCs but not OHCs.

### _Tg(DnRest)_ rescues IHCs and VHCs but fails to restore splicing regulation in SRRM3–SRRM4 double deficient mice

We tested whether _Srrm3_ expression is necessary for the _Tg(DnRest)_-dependent survival of IHCs and VHCs in _Srrm4^{bv/bv}_ mice. To reduce _Srrm3_ expression in _Tg(DnRest)_;_Srrm4^{bv/bv}_ mice, we took advantage of a recently produced gene trap allele of _Srrm3_ (_Srrm3^{gt}_) (Nakano et al, 2019). _Tg(DnRest)_;_Srrm3^{gt/gt}_;_Srrm4^{bv/bv}_ mice were generated and used for qRT-PCR and histological analyses. qRT-PCR testing of inner ear samples from these and _Srrm3^{gt/gt}_

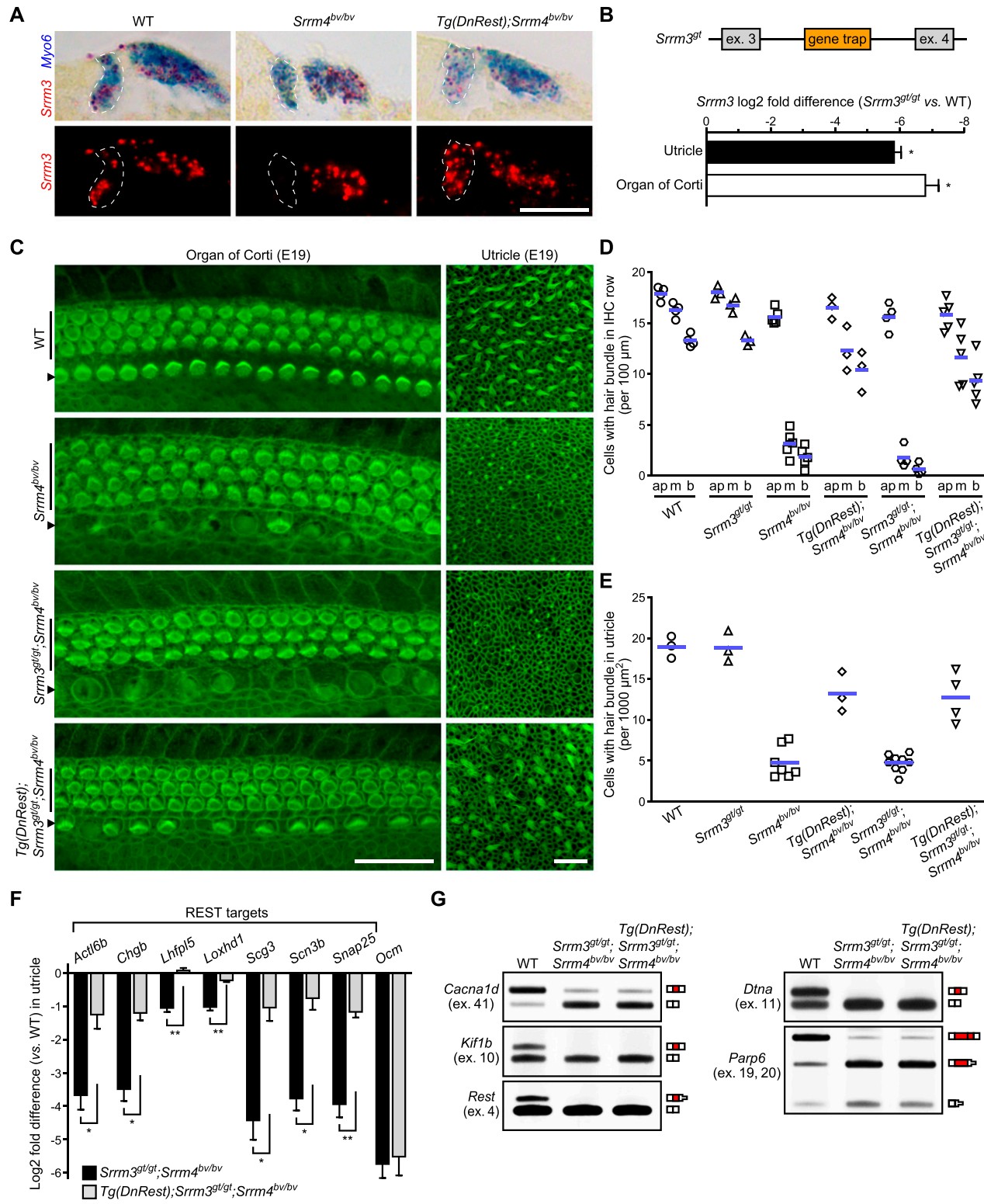

**Figure 3. SRRM3 is required for some of the rescue effects of *Tg(DnRest)* in *Srrm4^bv/bv* mice.**
**(A)** In situ hybridization of sections of the apical turn of the cochlea from mice of the indicated genotypes (P0), with probes complementary to *Srrm3* (red) and *Myo6* (blue) (n = 3 mice per genotype). Upper panels show bright-field images; lower panels show fluorescence of *Srrm3* probe. Inner hair cells (IHCs) are encircled by dashed lines. Scale bar, 20 μm. **(B)** Gene trap mutagenesis of *Srrm3*. Top: Schematic of the gene trap insertion site in *Srrm3*. Exons 3 and 4 (gray boxes), introns (horizontal lines), and gene trap (orange box) are indicated. Bottom: Results of qRT-PCR testing of *Srrm3* expression in the utricle and organ of Corti of *Srrm3^gt/gt* versus WT mice at P1. Values are mean ± SEM (n = 3 mice per genotype, one-sample *t* test, theoretical mean = 0, false discovery rate–adjusted *P < 0.01). **(C)** F-actin staining of whole-mount

mice confirmed that the gene trap inhibited *Srrm3* expression effectively (Figs 3B and S3B). F-actin staining of cochlear and utricular samples revealed that fewer IHCs and VHCs degenerated in *Tg(DnRest);Srrm3^{gt/gt};Srrm4^{bv/bv}* mice than in either *Srrm3^{gt/gt};Srrm4^{bv/bv}* or *Srrm4^{bv/bv}* mice at E19 (Fig 3C–E). In *Srrm3^{gt/gt}* and WT mice, F-actin staining did not reveal any HC degeneration at E19 (Figs 3C–E and S3C). At postnatal times, the HCs of *Srrm3^{gt/gt};Srrm4^{bv/bv}* mice could not be analyzed in vivo because this genotype is associated with early lethality due to an inability to breathe (Nakano et al, 2019). Together, these data indicate that SRRM3 is not required for the *Tg(DnRest)*-dependent rescue of IHCs and VHCs in the *Srrm4^{bv/bv}* mouse until E19 or later.

We next tested whether SRRM3 is required for the *Tg(DnRest)*-dependent rescue of utricular gene expression in *Srrm4^{bv/bv}* mice. qRT-PCR–based analysis of the utricular expression of seven REST target genes and *Ocm* at E16.5 revealed that only the former were expressed at higher levels in *Tg(DnRest);Srrm3^{gt/gt};Srrm4^{bv/bv}* than *Srrm3^{gt/gt};Srrm4^{bv/bv}* mice (Fig 3F). We used RT-PCR on these utricular samples to test the alternative splicing of six SRRM4-regulated exons that were also analyzed in *Tg(DnRest);Srrm4^{bv/bv}* mice in Fig 2B. We found that all were spliced into a lower percentage of their respective mRNAs in *Srrm3^{gt/gt};Srrm4^{bv/bv}* and *Tg(DnRest);Srrm3^{gt/gt};Srrm4^{bv/bv}* mice than in WT and *Tg(DnRest);Srrm4^{bv/bv}* mice (Figs 2B, 3G, and S3D). These data indicate that some, but not all, of the rescue effects of *Tg(DnRest)* require SRRM3 in *Srrm4^{bv/bv}* mice. Specifically, the rescue of splicing regulation and *Ocm* expression requires SRRM3, whereas the rescue of the expression of REST target genes does not.

### IHCs survive, and OHCs degenerate, in organ cultures derived from *Tg(DnRest);Srrm3^{gt/gt};Srrm4^{bv/bv}* mice

Given that the *Srrm3^{gt/gt};Srrm4^{bv/bv}* mice die immediately after birth (Nakano et al, 2019), we used organ of Corti cultures to test the effect of *Tg(DnRest)* on the survival of SRRM3–SRRM4–deficient IHCs after E19. These cultures were derived from E19 *Tg(DnRest);Srrm3^{gt/gt};Srrm4^{bv/bv}* and control mice (i.e., WT, *Srrm3^{gt/gt}*, *Srrm4^{bv/bv}*, *Tg[DnRest];Srrm4^{bv/bv}*, and *Srrm3^{gt/gt};Srrm4^{bv/bv}*), and F-actin–rich structures were visualized on day in vitro (DIV) 9 using fluorescently labeled phalloidin. The F-actin staining revealed that many stereociliary bundles were present in the IHC row in the cultures from WT, *Srrm3^{gt/gt}*, *Tg(DnRest);Srrm4^{bv/bv}*, and *Tg(DnRest);Srrm3^{gt/gt};Srrm4^{bv/bv}* mice, but very few were present in those from *Srrm4^{bv/bv}* and *Srrm3^{gt/gt};Srrm4^{bv/bv}* mice (Figs 4A and B and S4A). Thus, IHCs were rescued by *Tg(DnRest)* in the *Srrm3^{gt/gt};Srrm4^{bv/bv}* organ of Corti cultures. These data suggest that the expression of *Srrm3* is

not needed for the *Tg(DnRest)*-dependent rescue of IHCs in *Srrm4^{bv/bv}* mice during early postnatal development.

In the case of OHCs, analysis of the same DIV9 samples demonstrated that nearly all such cells in organ of Corti cultures from *Srrm3^{gt/gt};Srrm4^{bv/bv}* and *Tg(DnRest);Srrm3^{gt/gt};Srrm4^{bv/bv}* mice lost their stereocilia or were replaced by non-sensory epithelia (Fig 4A and B). The extent of OHC loss was similar in the *Srrm3^{gt/gt};Srrm4^{bv/bv}* and *Tg(DnRest);Srrm3^{gt/gt};Srrm4^{bv/bv}* cultures (Fig 4B), consistent with the fact that expression of *Tg(DnRest)* in OHCs was minimal (Fig 1C). In contrast, many OHCs in organ of Corti cultures derived from WT, *Srrm3^{gt/gt}*, *Tg(DnRest);Srrm4^{bv/bv}*, and *Srrm4^{bv/bv}* mice contained stereocilia bundles (Figs 4A and S4A). Thus, double deficiency of SRRM3 and SRRM4 causes degeneration of OHCs in organ of Corti cultures.

### REST inactivation and *Srrm3* expression do not require SRRM4 in OHCs

We next tested the effect of the SRRM3–SRRM4 double deficiency on the expression of REST target genes in OHCs. Given that REST target genes are expressed in both the OHCs and IHCs (though not the non-HCs) of WT mice (Scheffer et al, 2015; Li et al, 2018; Liu et al, 2018; Kolla et al, 2020), we collected IHC-free clusters of OHCs and the intercalated non-HCs for mRNA analysis (Fig S4B) (see also the Materials and Methods section). The cell clusters were isolated from DIV5 organ of Corti cultures that had been derived from E19 WT, *Srrm3^{gt/gt}*, *Srrm4^{bv/bv}*, and *Srrm3^{gt/gt};Srrm4^{bv/bv}* mice. We chose DIV5 as the time of cell collection because the OHCs of *Srrm3^{gt/gt};Srrm4^{bv/bv}* mice do not degenerate until then (data not shown). The isolated cell clusters were analyzed by qRT-PCR to determine expression levels of the HC-specific mRNA *Myo6*, the IHC-specific mRNA *Fgf8*, and the REST target genes that were also tested in Fig 2A. *Fgf8*-expressing samples were excluded from further analysis, and REST target gene expression was normalized to that of *Myo6*. The normalized data revealed that the expression of nearly all tested REST target genes was lower in *Srrm3^{gt/gt};Srrm4^{bv/bv}* samples than in WT, *Srrm3^{gt/gt}*, and *Srrm4^{bv/bv}* samples (Fig 4C). The qRT-PCR data also demonstrated that *Srrm3* expression did not differ between the *Srrm4^{bv/bv}* and WT samples, consistent with our in situ hybridization analysis of OHCs in *Srrm4^{bv/bv}* and WT mice (Fig 3A). Together, these data indicate that SRRM3 and SRRM4 act redundantly in maturing OHCs to up-regulate the expression of REST target genes.

The OHC-containing cell clusters were further analyzed by RT-PCR for alternative splicing of the six exons that had been tested in the experiments shown in Figs 2B and 3G. This analysis revealed that all six exons were spliced into a much lower percentage of their

---

preparations of middle-turn region of organs of Corti (left column) and utricles (right column) from E19 mice of the indicated genotypes (n is shown in Fig 3D and E). The rows of outer hair cells (vertical lines) and IHCs (arrowheads) are indicated. Scale bars, 20 μm. **(D, E)** Numbers of cells with stereocilia (hair) bundles in the IHC row of organ of Corti (D) and in the utricle (E) of E19 mice of the indicated genotypes. IHC data (D) are shown for the apical (ap), middle (m), and basal (b) turn regions of the organ of Corti. Each symbol represents value for a single mouse; blue lines indicate means. **(F)** qRT-PCR data for utricular expression of the indicated genes in E16.5 *Srrm3^{gt/gt};Srrm4^{bv/bv}* and *Tg(DnRest);Srrm3^{gt/gt};Srrm4^{bv/bv}* mice versus WT mice. Values are mean ± SEM (n = 4–5 mice per genotype, Welch's unpaired *t* test, false discovery rate–adjusted *P < 0.05, **P < 0.01). **(G)** RT-PCR analysis of alternative splicing of the indicated exons (ex.) in the utricles of E16.5 WT, *Srrm3^{gt/gt};Srrm4^{bv/bv}*, and *Tg(DnRest);Srrm3^{gt/gt};Srrm4^{bv/bv}* mice. Exon composition of RT-PCR products is depicted next to gel images (n = 3 mice per genotype). Red boxes represent the tested alternative exons; white boxes represent constitutive exons complementary to the RT-PCR primers; reduction in box height indicates that a stop codon is present. Source data are available for this figure.

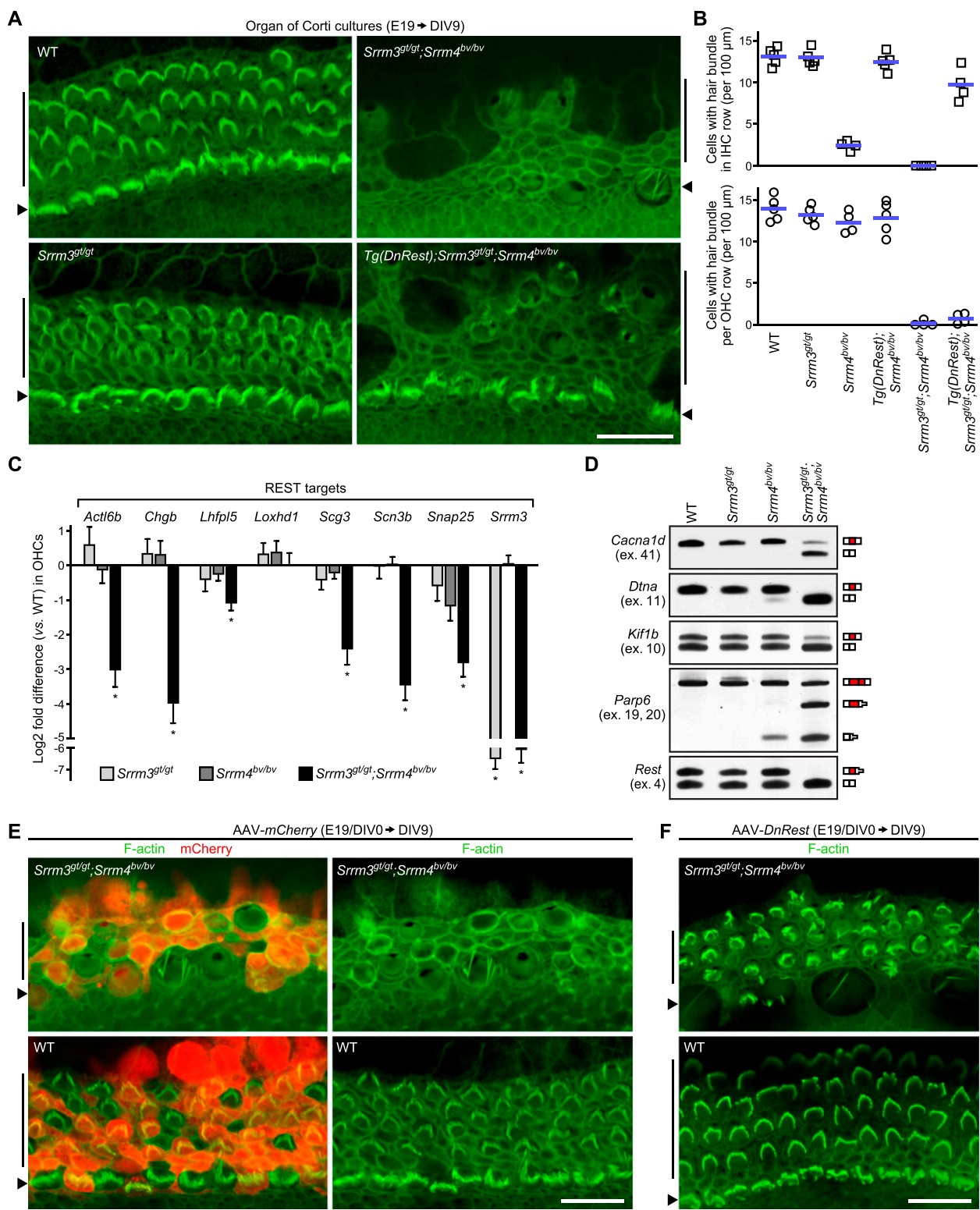

**Figure 4. SRRM3-dependent inactivation of REST prevents outer hair cell (OHC) loss in *Srrm4^bv/bv*^ mice.**
**(A)** F-actin–stained DIV9 organ of Corti cultures derived from E19 mice of the indicated genotypes (n is shown in Fig 4B). Positions of inner hair cells (IHCs) (arrowheads) and OHCs (vertical lines) are indicated next to images. Scale bar, 20 μm. **(B)** Counts of IHCs (top) and OHCs (bottom) with stereocilia (hair) bundles in DIV9 organ of Corti cultures derived from E19 mice of the indicated genotypes. Each symbol represents value for a single culture (one culture per mouse); blue lines indicate means. **(C)** Expression levels of the indicated REST target genes as determined by qRT-PCR in OHC-containing cell clusters isolated from DIV5 organ of Corti cultures of the indicated genotypes. Values are mean ± SEM (n = 3–4 mice per genotype, one-sample *t* test, theoretical mean = 0, false discovery rate–adjusted *P* < 0.05). **(D)** RT-PCR

respective mRNAs in OHC-containing cell clusters from *Srrm3^{gt/gt}*; *Srrm4^{bv/bv}* versus WT, *Srrm3^{gt/gt}*, and *Srrm4^{bv/bv}* cultures (Figs 4D and S4C). In OHC-containing cell clusters from WT versus *Srrm3^{gt/gt}* and *Srrm4^{bv/bv}* cultures, only minimal differences in splicing of the six exons were detected. Thus, SRRM3 and SRRM4 act redundantly in maturing OHCs to regulate the alternative splicing of several exons, including *Rest* exon 4.

### A DnREST-encoding virus rescues the OHCs of *Srrm3^{gt/gt}*;*Srrm4^{bv/bv}* mice in organ of Corti cultures

We next tested whether forced suppression of REST activity is sufficient to rescue the OHCs of *Srrm3^{gt/gt}*;*Srrm4^{bv/bv}* mice. Given that the expression of *Tg*(*DnRest*) in OHCs is minimal (Fig 1C), we used adeno-associated virus (AAV) (2.7m8 serotype [Isgrig et al, 2019]) to deliver a DnREST-encoding gene to OHCs in organ of Corti cultures. Specifically, organ of Corti cultures from E19 *Srrm3^{gt/gt}*; *Srrm4^{bv/bv}* and WT mice were transduced with AAV-*DnRest* and AAV-*mCherry* (control) at DIV0, and the degeneration of OHCs was evaluated based on F-actin staining of stereocilia bundles at DIV9. The F-actin staining revealed that in the *Srrm3^{gt/gt}*;*Srrm4^{bv/bv}* cultures, many more stereocilia bundles were present in the OHC rows after transduction with AAV-*DnRest* versus AAV-*mCherry* (Figs 4E and F and S4D and E). In WT cultures, such a difference was not apparent (Figs 4E and S4D and E). Thus, AAV-*DnRest* rescued the OHCs in *Srrm3^{gt/gt}*;*Srrm4^{bv/bv}* organ of Corti cultures until at least DIV9. These data indicate that SRRM3–SRRM4 double deficiency causes OHC death, and that it does so primarily because REST is not inactivated.

### *Rest* promoter activity is transiently repressed in maturing OHCs

We previously reported that at birth *Rest* promoter activity is lower in OHCs than IHCs (Nakano et al, 2018). To refine the window of *Rest* promoter down-regulation in perinatal OHCs, we tested *Rest* promoter activity on 7 d, from E16 to P9, using a mouse line that harbors a *Rest promoter-EGFP* transgene (*Tg*[*Rest pro-EGFP*]) (Gong et al, 2003). This transgene contains an EGFP-encoding sequence downstream of the *Rest* start codon, within a 160 kb BAC insert (Fig 5A). Activity of the transgenic *Rest* promoter in cochlear sections was evaluated based on the intensity of EGFP immunofluorescence, and HCs in the same sections were identified based on MYO7A immunofluorescence. This analysis revealed that the EGFP signal was transiently reduced in both OHCs and IHCs during maturation, but the magnitude and duration of the reduction in EGFP signal were much greater in OHCs than IHCs (Fig 5B and C).

Next, we tested expression of the endogenous *Rest* gene in perinatal OHCs and IHCs using duplex in situ hybridization. Cochlear sections from P0 WT mice were hybridized with probes complementary to the *Rest* and *Myo6* mRNAs (Fig 5D). Semiquantitative evaluation of the *Rest* signals in *Myo6*-labeled cells confirmed that *Rest* expression was much lower in OHCs than IHCs at P0 (Fig S5A). Together, these data indicate that the repression of *Rest* transcription during development is much greater in OHCs than IHCs.

Having tested *Rest* promoter activity in maturing HCs of the hearing organ, we also tested it in maturing HCs of the utricle. Cross sections of utricles from *Tg*(*Rest pro-EGFP*) mice were prepared on 7 d, from E16 to P9, and immunostained with anti-EGFP and anti-MYO7A antibodies (Fig S5B). Quantification of the intensity of EGFP immunofluorescence revealed that on each of these days the EGFP expression was only 40–50% lower in VHCs than in neighboring non-HCs (Fig S5C). In situ hybridization-based testing of the expression of endogenous *Rest* in the utricles of P0 mice confirmed that *Rest* was transcribed in maturing VHCs (Fig S5D). These data indicate that the *Rest* promoter is active in perinatal VHCs. Collectively, our data on *Rest* promoter activity in the developing inner ear demonstrate that the patterns of strong perinatal down-regulation of *Rest* transcription and HC survival in *Srrm4^{bv/bv}* mice are similar.

### Overexpression of REST represses *Srrm3* in maturing OHCs

Given that *Srrm3* is a target gene of REST, the transcriptional down-regulation of *Rest* in perinatal OHCs might be necessary for the SRRM4-independent expression of *Srrm3*. Alternatively, an as yet unknown transcriptional activator of *Srrm3* may predominate over REST in regulating *Srrm3*. We speculated that if the former mechanism is key to the SRRM4-independent up-regulation of *Srrm3*, overexpression of REST in maturing OHCs would strongly reduce the transcription of *Srrm3*. In contrast, if the latter mechanism is predominant, overexpression of REST would not change (or change only slightly) the transcription of *Srrm3*. To express REST in maturing OHCs, organ of Corti cultures from E17 WT mice were transduced on DIV0 with AAVs that encode full-length REST (AAV-*Rest*). Control samples were generated by transducing the organ of Corti cultures with AAV-*mCherry*. OHC-containing cell clusters were isolated from the transduced cultures on DIV5, which corresponds to P2–P3 in vivo, and the expression levels of *Srrm3* and *Srrm4* were tested using qRT-PCR. *Srrm4* was included in this analysis because the transcriptional activator ATOH1 had been suggested to predominate over REST in regulating *Srrm4* in maturing HCs (Nakano et al, 2018). The qRT-PCR data revealed that *Srrm3* expression was 68% ± 7% lower in OHC-containing cell clusters from AAV-*Rest*–transduced versus AAV-*mCherry*–transduced cultures (Fig 5E). In contrast, *Srrm4* expression was similar in the two sample groups (Fig 5E). The 68% ± 7% reduction in *Srrm3* expression in AAV-*Rest*–transduced cultures is substantial because the efficiency

---

analysis of alternative splicing of the indicated exons in OHC-containing cell clusters isolated from DIV5 organ cultures of the indicated genotypes (n = 3 mice per genotype). Exon composition of RT-PCR products is depicted next to gel images. Red boxes represent the tested alternative exons; white boxes represent constitutive exons complementary to the RT-PCR primers; reduction in box height indicates that a stop codon is present. **(E, F)** F-actin–stained (green) and mCherry-immunostained (red) DIV9 organ of Corti cultures derived from E19 mice of the indicated genotypes. **(E, F)** The cultures were incubated with AAV-*mCherry* (E) or AAV-*DnRest* (F) on DIV0 (n = 4–5 cultures for each mouse genotype–AAV genotype combination). Positions of OHCs (vertical lines) and IHCs (arrowheads) are indicated next to images. Scale bars, 20 μm.

Source data are available for this figure.

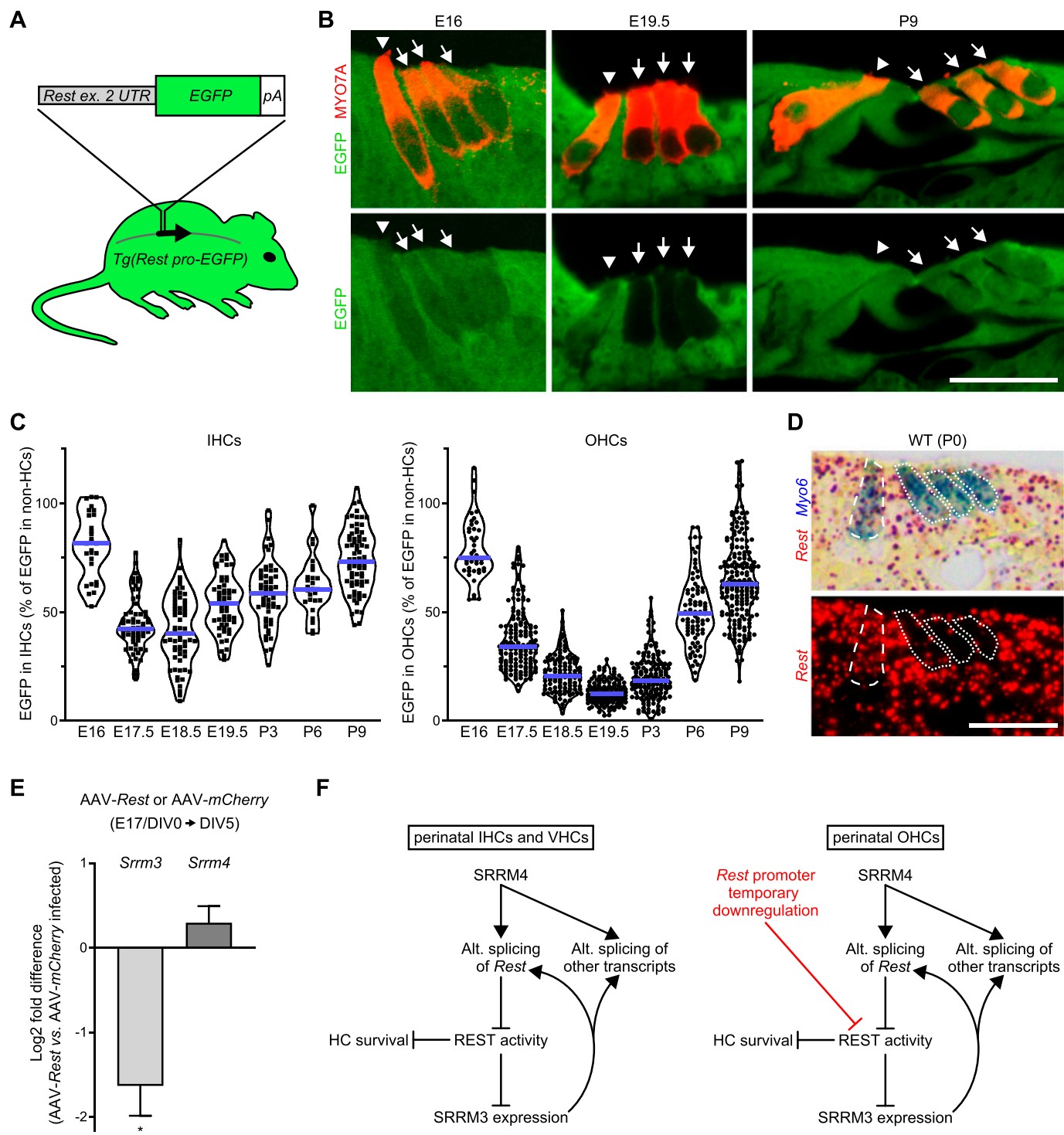

**Figure 5. Transient down-regulation of *Rest* transcription is associated with SRRM4-independent expression of *Srrm3* in maturing outer hair cells (OHCs).**
**(A)** Schematic of *Tg*(*Rest pro-EGFP*). Top: Organization of DNA elements inserted into mouse *Rest* in a bacterial artificial chromosome to create *Tg*(*Rest pro-EGFP*). Rectangles indicate UTR in exon (ex.) 2 of *Rest* (gray), EGFP-encoding sequence (green), and polyadenylation signal (white). Bottom: Position of modified exon 2 (vertical lines) in *Rest* (arrow), within the bacterial artificial chromosome (curved gray line) that was used for the production of *Tg*(*Rest pro-EGFP*) mice. **(B)** EGFP (green) and MYO7A (red) immunofluorescence in cochlear sections from *Tg*(*Rest pro-EGFP*) mice at E16, E19.5, and P9 (n = 2–3 mice per time point). Inner hair cells (IHCs) (arrowheads) and OHCs (arrows) are indicated. Scale bar, 20 μm. **(C)** Violin plots of *Tg*(*Rest pro-EGFP*) expression in IHCs (left graph) and OHCs (right graph) at the indicated times, as determined by EGFP immunofluorescence. EGFP signal was quantified as the percentage of that in non-hair cells in the same sections. Each symbol represents value for an individual cell (n = 2–3 mice per time point). Blue lines indicate medians. **(D)** In situ hybridization of a cochlear section from a P0 WT mouse, with probes complementary to *Rest* (red) and *Myo6* (blue) (n = 2 mice). Upper panel shows bright-field image; lower panel shows fluorescence of *Rest* probe. IHC is encircled by dashed line; OHCs are encircled by dotted lines. Scale bar, 20 μm. **(E)** Expression of *Srrm3* and *Srrm4* in OHC-containing regions from AAV-*Rest*-transduced versus AAV-*mCherry*–transduced WT

of transduction was 77% ± 4% for OHCs, as determined based on the number of red fluorescent OHCs in the AAV-*mCherry*–incubated cultures (Fig S5E). The AAV-*Rest*–dependent reduction in *Srrm3* expression was not caused by the death of OHCs because F-actin staining of AAV-*Rest*–incubated cultures revealed no loss of OHCs at DIV5 (Fig S5E). These data support the notion that down-regulation of transcription of *Rest* is necessary for the SRRM4-independent expression of *Srrm3* in perinatal OHCs.

## Discussion

In this study, we show that the primary and essential role of SRRM4 in the hearing and balance organs is REST inactivation. This conclusion is supported by the *Tg*(*DnRest*)-dependent long-term rescue of hearing, balancing, IHCs, and VHCs in *Srrm4^{bv/bv}* mice (Fig 1). In addition, our molecular analysis of *Tg*(*DnRest*);*Srrm4^{bv/bv}* mice reveals that OHCs differ from IHCs and VHCs with respect to the hierarchy of regulatory interactions among SRRM4, REST, and SRRM3 (Fig 5F). Specifically, in maturing OHCs, SRRM3 and SRRM4 act redundantly to inactivate REST, and SRRM4-dependent REST in-activation is not necessary for derepression of the REST target gene *Srrm3*, probably because *Rest* is transcriptionally silenced for a brief period around birth. In maturing IHCs and VHCs, however, SRRM3 and SRRM4 do not act redundantly because *Srrm3* is not derepressed in the absence of SRRM4. In these cells, transcriptional down-regulation of *Rest* is weak and SRRM4-dependent inactivation of REST is critical for the derepression of *Srrm3*.

Our data also suggest that the SRRM3 protein maintains the expression of its own gene in OHCs independently of SRRM4, and that it does so by inactivating REST. This model of SRRM3 regulation is based on four observations: SRRM3 efficiently inactivates REST in OHCs of *Srrm4^{bv/bv}* mice (Fig 4C and D); *Srrm3* mRNA levels are similar in the OHCs of *Srrm4^{bv/bv}* and WT mice (Fig 4C); *Srrm3* is a target gene of REST (Lee et al, 2015); and *Srrm3* expression is abnormally low in AAV-*Rest*-transduced OHCs of WT mice (Fig 5E). Determining whether SRRM3 can maintain the expression of *Srrm3* in VHCs and IHCs independently of SRRM4 will require postnatal deletion of a conditional *Srrm4* allele because SRRM4 is necessary for the "initial" up-regulation of SRRM3 in these cells.

Our DnREST-dependent rescue of both vestibular and cochlear HCs of *Srrm3^{gt/gt}*;*Srrm4^{bv/bv}* mice suggests that the primary role of combined SRRM3–SRRM4 activity in relation to HC survival is also REST inactivation. This notion is further supported by the similarities in morphological and molecular defects that characterize the inner ear in both *Srrm3^{gt/gt}*;*Srrm4^{bv/bv}* and *Rest^{+/ΔEx4}* mice (Figs 3 and 4) (Nakano et al, 2018). Specifically, the IHCs and VHCs start to degenerate at ~E17, and most of these cells lose their stereocilia bundles by P0. The timing of onset of OHC loss is also similar in the two mouse lines; OHC degeneration starts at ~P4 in *Rest^{+/ΔEx4}* mice

and at ~DIV5 in organ of Corti cultures derived from E19 *Srrm3^{gt/gt}*; *Srrm4^{bv/bv}* mice. In addition, the inner ears of mice of both genotypes are characterized by abnormally low expression of several REST target genes, as well as by abnormally low inclusion of *Rest* exon 4 in the mature mRNA. Together, these data suggest that the roles of SRRM3 and SRRM4 in *Rest* alternative splicing are more important for perinatal HC survival than are their effects on the splicing of other pre-mRNAs.

We do not propose that exon 4 of *Rest* is the only SRRM3–SRRM4–regulated exon required for the long-term survival and function of HCs. Our data show that DnREST rescued the alternative splicing of not only *Rest* but several other pre-mRNAs in the VHCs of *Tg*(*DnRest*);*Srrm4^{bv/bv}* mice by up-regulating *Srrm3* (Figs 2B and 3G). Given that the VHCs of these mice were functional until at least P120 (Fig S1G), we suggest that DnREST rescues the splicing of all SRRM4-regulated alternative exons that are essential for the function and long-term survival of VHCs.

Whereas the splicing of *Rest* exon 4 into the mature mRNA is known to inactivate the encoded transcriptional repressor, the effects of most other SRRM3–SRRM4–regulated exons on the functions of encoded proteins are unknown. An exception is the SRRM3–SRRM4–regulated splicing of exons 19 and 20 of *Parp6*. These two exons need to be spliced into the *Parp6* mRNA to maintain the translational reading frame (Fig 2B), and the skipping of either results in a premature termination codon over 150 bases upstream of the last exon–exon junction. Termination codons at such positions prevent the expression of the encoded protein and cause nonsense-mediated mRNA decay (Popp & Maquat, 2013). Thus, PARP6 expression in HCs is likely abnormally low in the absence of SRRM3–SRRM4 activity. The long-term effects of abnormally low PARP6 expression on HCs are currently unpredictable because PARP6 inactivating mutations have not been described in any species. The vast majority of other SRRM3–SRRM4–regulated exons do not change the translational reading frame, and most of these exons modify the encoded protein by only a few amino acids (Quesnel-Vallières et al, 2015; Nakano et al, 2019). Some of these seemingly minor changes in protein structure have been suggested to modulate the strength of protein–protein interactions (Irimia et al, 2014; Yang et al, 2016). Future studies will determine whether the SRRM3-dependent splicing of frame-preserving exons contributes to the long-term rescue of VHCs and IHCs in *Tg*(*DnRest*); *Srrm4^{bv/bv}* mice.

In light of the fact that the *Srrm4^{bv/bv}* genotype is associated with many defects in alternative splicing in the inner ear, the nearly complete rescue of inner ear function in *Srrm4^{bv/bv}* mice through the correction of a single regulatory defect (i.e., the defect in REST inactivation) is notable. Several splicing factors other than SRRM4—ESPR1, SFSWAP, and NOVA2—are important for the regulation of pre-mRNA splicing in the ear (Moayedi et al, 2014; Saito et al, 2016; Rohacek et al, 2017). Inactivation of the *Espr1* gene in mice

organ of Corti cultures as determined by qRT-PCR. The timing of organ culture preparation (E17), AAV infection (DIV0), and isolation of OHC-containing cell clusters from the cultures (DIV5) are shown above the graph. Values are mean ± SEM (n = 4 cultures per AAV genotype, one-sample *t* test, theoretical mean = 0, false discovery rate–adjusted ∗*P* = 0.0042). **(F)** Schematics of identified regulatory interactions among SRRM4, REST, and SRRM3 in perinatal IHCs (left diagram), vestibular hair cells (VHCs, left diagram), and OHCs (right diagram). Arrows represent activating interactions; T-shaped lines represent inhibitory interactions. Red text and line indicate a difference in *Rest* regulation in OHCs versus IHCs and vestibular hair cells.

is associated with aberrant cochlear morphogenesis, a delay in HC differentiation, and dysregulation of development of the stria vascularis; the last of these defects is caused by the production of a mesenchymal splice form of *Fgfr2* in epithelial cells of the inner ear (Rohacek et al, 2017). Disruption of the *Sfswap* gene in mice is associated with hearing impairment, a balance defect, and abnormal patterning of the cochlear sensory epithelium. Although the *Sfswap* mutant mice have not been analyzed for alterations in pre-mRNA splicing, their cochlear pathology suggests that Notch signaling is impaired (Moayedi et al, 2014). NOVA2 is necessary for afferent innervation of the cochlea, and pre-mRNAs of many axon guidance-related proteins contain NOVA2-regulated exons (Saito et al, 2016). Together, these studies indicate that an understanding of the molecular program underlying cochlear development will require identification of the critical exon targets of several splicing regulators.

Another essential step toward understanding the role of alternative splicing in cochlear development will be the identification of splicing regulators for numerous alternative exons that are known to be required for HC function. For example, the inclusion of alternatively spliced exons of *Pcdh15*, *Myo15*, and *Whrn* in their respective mRNAs is critical for proper organization of stereocilia (Webb et al, 2011; Fang et al, 2015; Ebrahim et al, 2016). Not only the inclusion, but also the skipping, of specific alternative exons during pre-mRNA splicing is important for hearing. Skipping of the longest coding exon of *Xirp2* produces a splice form required for maintaining proper stereocilium morphology (Francis et al, 2015), and skipping of a polyadenylation site–containing region of exon 8 of *Triobp* (which is dependent on alternative splicing) is necessary for maintaining stereocilia rootlets (Katsuno et al, 2019). Several other alternatively spliced exons have been identified in transcripts that encode proteins essential for hearing, although the physiological importance of these exons has not been tested in vivo (Kollmar et al, 1997; Rosenblatt et al, 1997; Jones et al, 1999; Di Palma et al, 2001; Chicka & Strehler, 2003; Beisel et al, 2005, 2007; Grati et al, 2006; Hill et al, 2006; Liang et al, 2006; Riazuddin et al, 2006; Shen et al, 2006; Alagramam et al, 2007; Rocha-Sanchez et al, 2007; Miranda-Rottmann et al, 2010; Chen et al, 2011; Sakai et al, 2011; Västinsalo et al, 2011; Narui & Sotomayor, 2018; Ranum et al, 2019). Given that these alternative exons have not been identified as targets of SRRM3- and SRRM4-dependent splicing regulation, we suggest that currently unidentified splicing factors regulate many alternative splicing events that are physiologically important in HCs. Our study and others discussed here demonstrate the importance of testing the functional effects of individual alternative exons in vivo, as these analyses have the potential to reveal complex regulatory interactions that are critical for HC development and hearing.

# Materials and Methods

### Mice

All mouse procedures were approved by the University of Iowa Institutional Animal Care and Use Committee. To produce *Tg*(*DnRest*), the *Myo7a*-harboring BAC RP24-116K5 (purchased from Children's Hospital Oakland) was modified in *Escherichia coli* using homologous recombination-based DNA insertion. The DNA insert was produced by fusing a Flag-DnREST–encoding sequence, a polyadenylation site, a kanamycin resistance gene, and two 50-bp regions from *Myo7a* (i.e., homology arms in Table S1) in a series of overlap extension PCRs. This DNA insert was electroporated into *E. coli* that contained the RP24-116K5 BAC and the DNA recombinase–encoding pRed/ET plasmid (Gene Bridges). Recombination-positive colonies were isolated using kanamycin, and correct insertion of the Flag-DnREST-encoding sequence into the BAC was confirmed using Southern blotting (Fig S1A). To produce *Tg*(*DnRest*) mice, the Flag-DnREST–encoding BAC was microinjected into the pronuclei of fertilized oocytes at Taconic Biosciences. These cells were transferred into the oviducts of pseudopregnant females, and *Tg*(*DnRest*) pups were identified based on PCR testing of tail DNA (see primers in Table S2). The *Tg*(*Rest pro-EGFP*) and *Srrm3*$^{gt/+}$ mouse lines were described previously (Gong et al, 2003; Nakano et al, 2019).

### Southern blotting of *Tg*(*DnRest*)-harboring BAC DNA

BAC DNA was isolated from bacteria using the Large-Construct Kit (QIAGEN) and incubated with NotI and XhoI restriction enzymes (New England Biolabs). Each restriction enzyme–incubated sample was loaded into two wells of an agarose gel and separated by size using a pulsed-field gel electrophoresis system (Bio-Rad). The gel was cut into two halves; one half was stained for DNA using SYBR Gold (Thermo Fisher Scientific) and the other half was blotted onto positively charged nylon membrane (Roche). This membrane was then hybridized with a digoxigenin-labeled probe generated from the DnREST-encoding sequence using the DIG DNA Labeling and Detection Kit (Roche). Finally, probe hybridization signals were detected using alkaline phosphate-conjugated anti-DIG antibody (Roche) and CDP-Star chemiluminescent substrate (Sigma-Aldrich).

### In situ hybridization

Mice were transcardially perfused with 4% PFA in PBS while under deep anesthesia. The inner ear was dissected from the skull of these mice, fixed with 4% PFA for 16 h, embedded into Tissue Freezing Medium (General Data Healthcare, Inc.), and cryosectioned. Type-6 probes specific for *Myo6* and type-1 probes specific for *Srrm3* and *Rest* were designed and synthesized at Thermo Fisher Scientific. These transcript-specific probes and secondary probes were hybridized to cryosections using the ViewRNA ISH Tissue 2-Plex Assay Kit (Thermo Fisher Scientific) according to the manufacturer's instructions.

### Cell lines and tissue culture

HEK293 cells (obtained from the American Type Culture Collection) were grown in DMEM/F-12 medium (Corning) supplemented with 10% fetal bovine serum (Atlanta Biologicals), penicillin (100 units/ml; Sigma-Aldrich), and streptomycin (100 mg/ml; Thermo Fisher Scientific). Organs of Corti were dissected from E17–P1 mice and placed on Matrigel-coated transwell plates (Corning). The organ cultures were maintained in neurobasal-A medium supplemented with B-27, N-2, 0.5 mM L-glutamine (all from Thermo Fisher

Scientific), and 100 U/ml penicillin (Sigma-Aldrich). This culture medium was replaced once every 3 d. Organ cultures and cell lines were grown at 37°C in the presence of 5% $CO_2$.

### OHCs collection, RNA extraction and RT-PCR analysis

Organ of Corti cultures were incubated with 5 $\mu$M FM1-43 (Thermo Fisher Scientific) in HBSS for ~15 s at room temperature to label HCs fluorescently. Immediately after this step, cultures were incubated with a mixture of thermolysin (0.2 mg/ml; Sigma-Aldrich) and DNase I (100 unit/ml; Worthington) in DMEM/F-12 for 5–6 min at 37°C. The enzymatic digestion was quenched with an equal volume of ice-cold DMEM/F-12 containing 10% FBS, and cultures were triturated to release OHC-containing cell clusters into the medium. OHC-containing cell clusters were selectively pipetted into low-binding barrier tips (MidSci) under a fluorescence microscope, and the cell clusters were transferred into a second DMEM/F-12-containing dish (ultra-low attachment culture dish; Corning) to wash away co-pipetted cell debris. Finally, the OHC-containing cell clusters were collected from the second dish by selective pipetting under a fluorescence microscope. RNA was extracted from the OHC-containing cell clusters and dissected vestibular maculas using the RNeasy Micro Kit (QIAGEN). RNA samples were reverse-transcribed using SuperScript IV (Thermo Fisher Scientific). The cDNA samples were analyzed using the Hot-StarTaq PCR Kit (QIAGEN) or the PerfeCTa SYBR Green FastMix (Quantabio). *Heatr3* was used as the reference mRNA in Fig S2A; in all other qRT-PCR experiments, *Myo6* was used as the reference mRNA. Relative mRNA expression values were determined using the $\Delta\Delta$CT method (Livak & Schmittgen, 2001). qRT-PCR assay efficiency was tested for each primer pair using threefold dilution series of mouse utricular cDNA samples as templates (Bustin et al, 2009). All qRT-PCR assays in this study had efficiencies between 89% and 113%. RT-PCR and qRT-PCR primers are shown in Table S2.

### Luciferase assay

HEK293 cells were co-transfected with a REST activity reporter construct (RE1-TK-pGL4.10), a transfection efficiency reporter construct (TK-pGL4.7Rluc), and a DnREST-encoding plasmid using Lipofectamine LTX and PLUS reagent (Thermo Fisher Scientific). The RE1-TK-pGL4.10 construct contained a REST-binding site (RE1) upstream of the thymidine kinase gene promoter (TK) and a firefly luciferase–encoding sequence; the TK-pGL4.7Rluc construct contained TK and a Renilla luciferase–encoding sequence; the DnREST-encoding plasmid contained the previously published sequence of *Flag-DnRest*, *DnRest*, or no insert (negative control) downstream of the CMV promoter in pcDNA3.1+ vector (Tapia-Ramírez et al, 1997; Nakano et al, 2019). The transfected HEK293 cell cultures were lysed 36 h after transfection, and activities of firefly and renilla luciferases in the lysates were measured using the Dual-Luciferase Reporter Assay (Promega) and a luminometer (Wallac 1420 Multi-label Counter; PerkinElmer Life Sciences).

### Hearing and balance tests

The ABR thresholds of mice were measured using a previously described open-field system (Swiderski et al, 2014) and broadband click stimuli. The ability of mice to balance was evaluated by measuring the time each mouse remained on a fixed horizontal rod (1.8 cm in diameter) following two training trials.

### Immunofluorescence and F-actin staining

Whole-mount preparations of cochlear and vestibular tissues were fixed in 4% PFA and incubated with an anti-MYO7A antibody (RRID: AB_10015251; Proteus Biosciences) or phalloidin-Alexa 488 (Thermo Fisher Scientific). The binding of anti-MYO7A antibody to the specimens was tested using an Alexa 594-conjugated secondary antibody (RRID: AB_141637; Thermo Fisher Scientific). *Tg*(*Rest pro-EGFP*) expression in the ear was tested by fixation of whole-mount preparations of cochlear and vestibular tissues from *Tg*(*Rest pro-EGFP*) mice and WT mice (negative control) in 4% PFA, incubation with an Alexa 488–conjugated anti-GFP antibody (RRID: AB_10003058; Novus Biologicals) and an Alexa 568–conjugated anti-MYO7A antibody (RRID: AB_10015251; Proteus Biosciences), embedding into Tissue Freezing Medium (General Data Healthcare, Inc.), cryosectioning, and confocal microscopy. The APEX Antibody Labeling Kit (Thermo Fisher Scientific) was used for the conjugation of Alexa dyes to the anti-GFP and anti-MYO7A antibodies. To examine the expression of *Tg*(*DnRest*) in HCs, inner ears from P1 mice were fixed in 4% PFA, embedded into paraffin, sectioned, and incubated with an anti-Flag antibody (RRID: AB_259529; Sigma-Aldrich) and an anti-MYO7A antibody (RRID: AB_10015251; Proteus Biosciences). The binding of anti-Flag and anti-MYO7A antibodies to the specimens was tested using secondary antibodies that were conjugated with Alexa 488 or Alexa 594. All images of fluorescently labeled tissues were acquired using an LSM 880 confocal microscope (Carl Zeiss) and analyzed using the ZEN lite 2012 software (Carl Zeiss).

### Production and use of AAV

AAV particles (AAV2.7m8 serotype, 0.5–3 × $10^{13}$ viral genome/ml) were produced in HEK293 cells at Vigene Biosciences using the triple transfection method (Ayuso et al, 2010). The three plasmids included a helper plasmid, a shuttle plasmid, and a Rep/Cap plasmid that encodes capsid proteins and replicases (Dalkara et al, 2013) (plasmid #64839; Addgene). Shuttle plasmids were produced by subcloning the DnREST-encoding, mCherry-encoding, and REST-encoding sequences between the CMV promoter and a polyadenylation site (pA) in the pFBAAV-CMV-mcs-BgHpA vector (from Viral Vector Core, University of Iowa). To transduce HCs with AAV, organs of Corti were dissected from mice and mounted on transwell inserts (1.12 $cm^2$ area) in 1 ml neurobasal-A medium supplemented with B-27, N-2, L-glutamine (0.5 mM), penicillin (100 U/ml), and AAV ($10^{11}$ viral genome). The culture medium was removed 24 h later, and AAV-free medium was added to the transwell cultures.

### Statistical analysis

Data are presented as individual data points or mean ± SEM, as indicated in the corresponding figure legends. The statistical methods applied are described in the figure legends. Sample sizes were not determined a priori. All statistical analyses were performed

using GraphPad Prism version 8. Assumptions of normality were tested using the Shapiro–Wilk test, and assumptions of equal variance were determined with either the F-test (for two groups) or the Brown–Forsythe test (for three or more groups).

In cases where assumptions of normality and equal variance were not violated, parametric tests were used for further analysis of datasets: the one-sample *t* test was used to compare the mean of one group to a hypothetical value; the unpaired *t* test was used to compare the means of two groups; and one-way or two-way ANOVA was used (depending on the number of categorical independent variables) to compare the means of three or more groups. When ANOVA revealed significant findings, Dunnett's or Tukey's post hoc test was used. Dunnett's test was selected when the means of test groups were compared to the mean of a control group, and Tukey's test was selected when all possible pairs of means were compared.

In cases where the assumption of equal variance (but not normality) was violated, Welch's *t* test or Welch's ANOVA was used. Welch's *t* test was used to compare the means of two groups, and Welch's ANOVA was used to compare the means of three or more groups. When Welch's ANOVA revealed significant findings, Dunnett's T3 test was used to compare all possible pairs of means.

In cases where the assumption of normality (but not equal variance) was violated, non-parametric tests were used: the Mann–Whitney test was used to compare the medians of two groups; and the Kruskal–Wallis test was used to determine whether one sample in a group of three stochastically dominates another. When the Kruskal–Wallis test revealed significant findings, Dunn's post hoc test was used to compare test groups with a control group.

In cases where assumptions of both normality and equal variance were violated by ceiling and floor effects (Figs 1E, S1G, and S5A), nonparametric tests were used (i.e., Mann–Whitney test or Kruskal–Wallis test, depending on the number of compared groups). In the context of these figures, ceiling and floor effects increase the probability of type II error (false negative) but not type I error (false positive).

In figure panels where multiple one-sample or two-sample *t* tests were used, the Benjamini-Hochberg procedure was applied to limit the false discovery rate at 0.05. *P* < 0.05 is considered statistically significant. Where individual data points are not shown in the corresponding figure, the numbers of animals and biological replicates of assays (n) are reported in the legends.

## Supplementary Information

## Acknowledgements

We thank Dr. Chantal Allamargot and Tom Moninger for technical assistance, Drs. John Flannery and David Schaffer for providing the adeno-associated virus 2.7m8 Rep/Cap plasmid, the Viral Vector Core of the University of Iowa for providing adeno-associated virus shuttle vectors, and Dr. Christine Blaumueller for critical review of the manuscript. This project was supported by grants from the National Institute on Deafness and Other Communication Disorders (https://www.nih.gov/ R01DC014953 to B Bánfi) and the National Institute on Aging (https://www.nih.gov/ R01AG060504 to B Fritzsch). This work was also supported by the Iowa City Department of Veterans Affairs Medical Center, which provided access to research equipment (to B Bánfi). The funders had no role in study design, data collection and analysis, decision to publish, or preparation of the manuscript.

## Author Contributions

Y Nakano: conceptualization, resources, data curation, formal analysis, validation, investigation, visualization, methodology, project administration, and writing—original draft, review, and editing.
S Wiechert: data curation and investigation.
B Fritzsch: resources and writing—review and editing.
B Banfi: conceptualization, resources, data curation, formal analysis, supervision, funding acquisition, validation, investigation, visualization, methodology, project administration, and writing—original draft, review, and editing.

## Conflict of Interest Statement

The authors declare that they have no conflict of interest.

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
