## [Reviewer comments · Life Science Alliance]

Life Science Alliance

Inhibition of a transcriptional repressor rescues hearing in a splicing factor-deficient mouse

Yoko Nakano, Susan Wiechert, Bernd Fritsch, and Botond Banfi

DOI: <https://doi.org/10.26508/lsa.202000841>

Corresponding author(s): Botond Banfi, University of Iowa

Review Timeline:	Submission Date:	2020-07-05
	Editorial Decision:	2020-08-28
	Revision Received:	2020-10-08
	Editorial Decision:	2020-10-09
	Revision Received:	2020-10-09
	Accepted:	2020-10-12

Scientific Editor: Shachi Bhatt

Transaction Report:

August 28, 2020

Re: Life Science Alliance manuscript #LSA-2020-00841

Dr. Botond Banfi
University of Iowa
Anatomy and Cell Biology
431 Newton Road
Iowa City, Iowa 52242

Dear Dr. Banfi,

Thank you for submitting your manuscript entitled "Inhibition of a transcriptional repressor rescues hearing in a splicing factor-deficient mouse" to Life Science Alliance (LSA). The manuscript has been reviewed by the editors and outside referees (reviewer comments below). As you will see, the reviewers were very enthusiastic about the study and its potential impact, and have raised only minor concerns that should be addressed prior to further consideration of the manuscript at LSA. Therefore, although we are unable to publish the current version of the manuscript, we would kindly encourage you to submit a revised version that addresses the referees' concerns.

Most of the concerns raised by the reviewers can be addressed by text changes. The experimental data for concerns raised by Rev 3, with regards to figures 3b and 4 should only be included if readily available. We would be happy to discuss the individual revision points further with you should this be helpful.

When submitting the revision, please include a letter addressing the reviewers' comments point by point. The typical timeframe for revisions is three months. Please note that papers are generally considered through only one revision cycle, so strong support from the referees on the revised version is needed for acceptance. While you are revising your manuscript, please also attend to the below editorial points to help expedite the publication of your manuscript. Please direct any editorial questions to the journal office.

Thank you for considering LSA as an appropriate venue for your research. Please reach out to us if you have any questions about the revisions or resubmission.

Sincerely,

Shachi Bhatt, Ph.D.
Executive Editor
Life Science Alliance

B. MANUSCRIPT ORGANIZATION AND FORMATTING:

Reviewer #1 (Comments to the Authors (Required)):

This paper presents a very thorough analysis of the role of Rest, Srrm4 and Srrm3 in the development and maintenance of sensory hair cells in the mouse. There is a significant amount of new data with validation and cross-checking of findings, giving an in-depth interpretation of the molecular control of hair cell development summarised in figure 5F. The results are quite densely-presented but the discussion is clear and easy to read. The work brings together and clarifies the contributions of Rest and Srrm4 to sensory hair cell development and introduces the role of Srrm3 to the molecular network.

Minor points:

p7-8, "Tg(DnRest) rescues gene expression and alternative splicing in the utricles of Srrm4^{bv/bv} mice": Did the authors test the expression of these genes and exons in Tg(DnRest) alone (Fig 2A)? It would be worth establishing what the transgene alone does to expression levels.

P8, last sentence of middle paragraph: This conclusion is not a strong summary of the data presented in this section, and it rests on something that is not known (which is not the same as known not to). Could the authors make a clearer statement here?

p11, last sentence of middle paragraph: The authors state that "Given that the expression of Tg(DnRest) in OHCs is minimal (Fig. 1C), the degeneration of these cells in Srrm3gt/gt;Srrm4bv/bv versus Tg(DnRest);Srrm3gt/gt;Srrm4bv/bv cultures was not compared". However, the counts are plotted in figure 4B so it is an odd thing to say. What do the authors mean by this statement?

p14:-Is "heterologous" the right word here? This generally means expression of a sequence from a different species, rather than (I assume) expression of a gene in a different cell type. Do the authors mean over-expression?

Methods:

Numbers of mice are not reported for qualitative data (eg images, Fig1CFH, and splicing, Fig2B). Hearing tests suggests that clicks and pure-tone stimuli were used, but only click data are presented.

Figures:

Fig 1G/S1F: It would be better to put the counts from the middle turn in Fig 1G, rather than putting them on their own in the supplementary figure.

Fig 1D, E, G: It would help if the authors added the age to the figure panels and to Fig S1D, E, F to distinguish the difference between the two sets of plots.

Reviewer #2 (Comments to the Authors (Required)):

REST is known to be associated with hearing loss in humans and mice. This work comprises an elegant set of experiments to test whether the deafness in Bronx waltzer mice is due to REST in activation. The authors set out to ask several questions: The main question is whether inactivation of REST is the primary and essential role of SRRM4? Is the deafness of Bronx waltzer mice related to the lack of REST inactivation? Is there reciprocal regulation between SRRM3 and REST? And how do different hair cells, vestibular vs inner vs outer hair cells, differ in their regulation? These questions were answered well. Expression of REST in transgenic mice rescued deafness in the bv mice. Moreover, a study of the differential expression and damage to different hair cells was done, which is critical, as it has become clear that there are differences between the hair cells of the cochlea and vestibule. SRRM3 and SRRM4 expression was compared, and found to be redundant in outer hair cells, but each is essential in all other hair cells. The authors used transgenic mice, RT-PCR, in situ hybridization and organ of Corti cultures to demonstrate their findings. The data is strongly supportive for each main point of the paper. In summary, there are no issues to address.

Reviewer #3 (Comments to the Authors (Required)):

Nakano et al build on their excellent studies from 2012, 2018 and 2019, to further explore the role of the splicing factors SRRM4 and SRRM3 and REST in the development and maintenance of sensory hair cells. They had previously shown that one of the genes for which its splicing was affected in

the *Srrm4* *bv/bv* mice was the transcriptional repressor REST (2012 Nakano et al), but had not shown yet that aberrant REST splicing is the main driver of inner hair cell degeneration. They do this in this study. The paper starts with a surprisingly (near) complete rescue of hair cells and hearing function in *Srrm4* *bv/bv* mice that overexpress a dominant-negative form of REST (DN-REST), suggesting that failed inactivation of REST in *Srrm4* *bv/bv* IHCs is the main factor for its degeneration. The authors then go on to show that *Srrm3* provides redundancy in a rather complex fashion and that a transient, strong repression of REST expression in OHCs might underlie their survival in the *Srrm4* *bv/bv* mice.

After the initial finding describing the DN-REST-mediated rescue, the paper becomes challenging to follow, mainly due to the mind-twisting and inevitable difficulty (at least for this reader) of following the logic of double-negatives, and due to the complexity of the interactions between *Srrm4*, *Srrm3*, and REST (kudos to the authors for figuring this out!). Despite the rather difficult read, the main conclusions of the paper are well supported by the data, and the findings will prove to be significant for the hearing research field.

A few rather minor comments that might improve the manuscript:

- The logic of recurring double-negatives (inactivation of a transcriptional repressor etc) is difficult to follow. This is just a suggestion, but to set the readers on the right path at the very start, it would help to start the Introduction with a fool-proof sentence: instead of "REST is a transcriptional repressor of hundreds of genes that are expressed selectively in neurons and HCs, and its activity is downregulated dramatically in both neurons and HCs as they undergo differentiation " maybe write it this way: "REST is a transcriptional repressor of hundreds of genes that are expressed selectively in neurons and HCs. Downregulation of REST activity during differentiation releases the transcriptional suppression and activates the expression of these genes important for neuronal and HC identity (differentiation)"

- Rotarod performance does not equate balance performance. This reviewer is not asking for VORs or VsEPs, but the statements regarding balance should be made a bit more cautious. Please change "Thus, Tg(DnRest) rescues hearing and balance in the *Srrm4**bv/bv* mouse line" to "Thus, Tg(DnRest) rescues hearing and rotarod performance in the *Srrm4**bv/bv* mouse line

- The authors express the data often in this manner: This analysis revealed that 3-4 fold more IHCs survived in Tg(DnRest);*Srrm4**bv/bv* mice than in *Srrm4**bv/bv* littermates (Fig. 1G, Fig. S1F and S1G). I would just state the actual numbers (WT: XX +/- error ; Mutant: YY +/- error, with p-values for stat difference).

- In Fig. 1F, it looks like the rescue of IHCs in the Tg(DN) mice is not 100%. Statistical treatment of data presented in Figure 1G and I would clarify that.

- Fig 2A: Here, not only the 8 REST target genes are downregulated in utricular hair cells at E16, but also *Ocm*, which is not known to be a REST target. The easiest explanation would be that the downregulation of 8 + *Ocm* is due to hair cell loss, but the authors exclude that possibility by using E16.5 utricles where there is no obvious hair cell loss. But to really convincingly exclude that HC loss is a factor, it would be good to have one additional (hair cell) control (e.g. *Myo7a*, *Myo6*) where the expression remains unchanged in the *Srrm4* *bv/bv* mice.

- Here is another example where the writing could be improved to aid in the understanding of this rather complex content: The result in Fig 2B is surprising, because the Tg(DN) rescues splicing. It would help to explicitly state that this is an unexpected result. Something like: "Thus, Tg(DnRest) rescues the alternative splicing of several *SRRM4*-regulated exons in the *Srrm4**bv/bv* mouse. This

is surprising, given that REST is not known to regulate pre-mRNA splicing directly. Our data therefore suggest that DnREST rescues alternative splicing in *Srrm4*^{bv/bv} mice through an indirect mechanism. "

- In Fig 3B, it is not clear whether the gene trap completely abolishes *Srrm3*, or whether there is some residual expression.

- On page 9, the authors write: We tested whether *Srrm3* expression is necessary for the Tg(DnRest)-dependent survival of IHCs and VHCs in *Srrm4*^{bv/bv} mice. *Srrm3* expression was reduced by outcrossing the Tg(DnRest);*Srrm4*^{bv/bv} mouse line to a recently generated line that is heterozygous for a gene trap insertion in *Srrm3* (*Srrm3*^{+/gt}). This sounds like the authors are using *Srrm* ^{+/gt} mice (although they use the *Srrm* ^{gt/gt}). Just omit the sentence "*Srrm3* expression was reduced by outcrossing the Tg(DnRest);*Srrm4*^{bv/bv} mouse line to a recently generated line that is heterozygous for a gene trap insertion in *Srrm3* (*Srrm3*^{+/gt})." and write that you generated Tg(DnRest);*Srrm3* ^{gt/gt}; *Srrm* ^{bv/bv} mice.

- In Fig.4, the Tg(DnRest);*Srrm3* ^{gt/gt}; *Srrm* ^{bv/bv}, there is no survival of OHCs. But as shown in the original Boeda et al and recently in Li et al, 2020 (*Myosin-VIIa* is expressed in multiple isoforms and essential for tensioning the hair cell mechanotransduction complex), the *Myo7a* promoter does drive some expression in apical OHCs. This means there should be some rescue in the apical OHCs. Have the authors checked that?

- First sentence of last paragraph in page 11: We next tested the effect of the *SRRM3-SRRM4* double deficiency on expression of the REST target genes that we had found to be downregulated in *Srrm4*^{bv/bv} versus WT utricles (Fig. 2A). Was this supposed to say "in OHCs"?

Overall, this is a nice study that caps some of the open questions from the author's previous high-quality and high-profile work on *Srrm4/3* and REST. The study makes a significant contribution towards the growing realization of the importance of splice variants for hair cell biology.

Point by point response

Reviewer #1: This paper presents a very thorough analysis of the role of *Rest*, *Srrm4* and *Srrm3* in the development and maintenance of sensory hair cells in the mouse. There is a significant amount of new data with validation and cross-checking of findings, giving an in-depth interpretation of the molecular control of hair cell development summarised in figure 5F. The results are quite densely-presented but the discussion is clear and easy to read. The work brings together and clarifies the contributions of *Rest* and *Srrm4* to sensory hair cell development and introduces the role of *Srrm3* to the molecular network.

Minor points:

p7-8, "*Tg(DnRest)* rescues gene expression and alternative splicing in the utricles of *Srrm4^{bv/bv}* mice": Did the authors test the expression of these genes and exons in *Tg(DnRest)* alone (Fig 2A)? It would be worth establishing what the transgene alone does to expression levels.

The reviewer makes an excellent point because it is possible that *Tg(DnRest)* could cause premature or excessive upregulation of REST target genes in developing HCs. Moreover, these changes in gene expression could cause morphological and functional defects in the inner ear. We therefore looked for defects in HC morphology, hearing, and motor coordination in mice of the *Tg(DnRest)* genotype. These tests did not reveal any difference between *Tg(DnRest)* and control (WT) mice, and the data have been added to Fig. 1D and 1E, Fig. S1D and S1E. Based on these findings, we suggest that *Tg(DnRest)* is not damaging to HCs.

Given our finding that *Tg(DnRest)* did not cause detectable damage to HCs, we did not proceed to test *Tg(DnRest)* and WT mice for potential differences in pre-mRNA splicing and gene expression in the inner ear. We considered that even if splicing and gene expression differ to some extent in *Tg(DnRest)* versus WT mice, our data in Fig. 2 support the conclusion that *Tg(DnRest)* rescues the expression and alternative splicing of several transcripts in the *Srrm4^{bv/bv}* mouse utricle.

P8, last sentence of middle paragraph: This conclusion is not a strong summary of the data presented in this section, and it rests on something that is not known (which is not the same as known not to). Could the authors make a clearer statement here?

We have reworded the ending of this paragraph:

"Thus, *Tg(DnRest)* rescues the alternative splicing of several SRRM4-regulated exons in the utricle of *Srrm4^{bv/bv}* mice. This is surprising because DnREST and REST are not known to regulate pre-mRNA splicing directly. Given that DnREST derepresses the splicing factor-encoding gene *Srrm3* in the utricle of *Srrm4^{bv/bv}* mice (Fig. 2A), the ability of DnREST to rescue alternative splicing could be due to SRRM3."

p11, last sentence of middle paragraph: The authors state that "Given that the expression of *Tg(DnRest)* in OHCs is minimal (Fig. 1C), the degeneration of these cells in *Srrm3^{gt/gt};Srrm4^{bv/bv}* versus *Tg(DnRest);Srrm3^{gt/gt};Srrm4^{bv/bv}* cultures was not compared". However, the counts are plotted in figure 4B so it is an odd thing to say. What do the authors mean by this statement?

We reworded this sentence to better convey that *Tg(DnRest)* was not expected to affect OHC survival because *Tg(DnRest)* is not expressed in OHCs:

"The extent of OHC loss was similar in the *Srrm3^{gt/gt};Srrm4^{bv/bv}* and *Tg(DnRest);Srrm3^{gt/gt};Srrm4^{bv/bv}* cultures (Fig. 4B), consistent with the minimal expression of *Tg(DnRest)* in OHCs (Fig. 1C)."

p14:-Is "heterologous" the right word here? This generally means expression of a sequence from a different species, rather than (I assume) expression of a gene in a different cell type. Do the authors mean over-expression?

We agree with the reviewer that overexpression is a better word in this context. We changed 'heterologous expression' to 'overexpression' in the text.

Methods:

Numbers of mice are not reported for qualitative data (eg images, Fig1CFH, and splicing, Fig2B).

We included the number of mice for all data in the revised figure legends.

Hearing tests suggests that clicks and pure-tone stimuli were used, but only click data are presented.

We thank the reviewer for bringing this error to our attention. We have deleted the description of pure-tone hearing tests from the Materials and Methods section.

Figures:

Fig 1G/S1F: It would be better to put the counts from the middle turn in Fig 1G, rather than putting them on their own in the supplementary figure.

We agree. Fig. S1F has been incorporated into Fig. 1G. We made a similar change in Fig. 3D/S3D (i.e., Fig. S3D is now incorporated into Fig. 3D).

Fig 1D, E, G: It would help if the authors added the age to the figure panels and to Fig S1D, E, F to distinguish the difference between the two sets of plots.

We agree with the reviewer. We have added the age of the mice at the time of testing to these figure panels.

Reviewer #2: REST is known to be associated with hearing loss in humans and mice. This work comprises an elegant set of experiments to test whether the deafness in Bronx waltzer mice is due to REST in activation. The authors set out to ask several questions: The main question is whether inactivation of REST is the primary and essential role of SRRM4? Is the deafness of Bronx waltzer mice related to the lack of REST inactivation? Is there reciprocal regulation between SRRM3 and REST? And how do different hair cells, vestibular vs inner vs outer hair cells, differ in their regulation? These questions were answered well. Expression of REST in transgenic mice rescued deafness in the bv mice. Moreover, a study of the differential expression and damage to different hair cells was done, which is critical, as it has become clear that there are differences between the hair cells of the cochlea and vestibule. SRRM3 and SRRM4 expression was compared, and found to be redundant in outer hair cells, but each is essential in all other hair cells. The authors used transgenic mice, RT-PCR, in situ hybridization and organ of Corti cultures to demonstrate their findings. The data is strongly supportive for each main point of the paper. In summary, there are no issues to address.

We thank the reviewer for his/her comments.

Reviewer #3: Nakano et al build on their excellent studies from 2012, 2018 and 2019, to further explore the role of the splicing factors SRRM4 and SRRM3 and REST in the development and maintenance of sensory hair cells. They had previously shown that one of the genes for which its splicing was affected in

the *Srrm4* *bv/bv* mice was the transcriptional repressor REST (2012 Nakano et al), but had not shown yet that aberrant REST splicing is the main driver of inner hair cell degeneration. They do this in this study. The paper starts with a surprisingly (near) complete rescue of hair cells and hearing function in *Srrm4* *bv/bv* mice that overexpress a dominant-negative form of REST (DN-REST), suggesting that failed inactivation of REST in *Srrm4* *bv/bv* IHCs is the main factor for its degeneration. The authors then go on to show that *Srrm3* provides redundancy in a rather complex fashion and that a transient, strong repression of REST expression in OHCs might underlie their survival in the *Srrm4* *bv/bv* mice. After the initial finding describing the DN-REST-mediated rescue, the paper becomes challenging to follow, mainly due to the mind-twisting and inevitable difficulty (at least for this reader) of following the logic of double-negatives, and due to the complexity of the interactions between *Srrm4*, *Srrm3*, and REST (kudos to the authors for figuring this out!). Despite the rather difficult read, the main conclusions of the paper are well supported by the data, and the findings will prove to be significant for the hearing research field.

A few rather minor comments that might improve the manuscript:

- The logic of recurring double-negatives (inactivation of a transcriptional repressor etc) is difficult to follow. This is just a suggestion, but to set the readers on the right path at the very start, it would help to start the Introduction with a fool-proof sentence: instead of "REST is a transcriptional repressor of hundreds of genes that are expressed selectively in neurons and HCs, and its activity is downregulated dramatically in both neurons and HCs as they undergo differentiation " maybe write it this way: "REST is a transcriptional repressor of hundreds of genes that are expressed selectively in neurons and HCs. Downregulation of REST activity during differentiation releases the transcriptional suppression and activates the expression of these genes important for neuronal and HC identity (differentiation)"

We thank the reviewer for the suggested edits. We reworded the first few sentences of the Introduction:

"REST is a transcriptional repressor of hundreds of genes that are expressed selectively in mature neurons and HCs. These genes are derepressed in neurons and HCs during development because REST activity is downregulated dramatically in both cell types (Chong et al., 1995; Schoenherr and Anderson, 1995; Nakano et al., 2018)."

- Rotarod performance does not equate balance performance. This reviewer is not asking for VORs or VsEPs, but the statements regarding balance should be made a bit more cautious. Please change "Thus, *Tg(DnRest)* rescues hearing and balance in the *Srrm4**bv/bv* mouse line" to "Thus, *Tg(DnRest)* rescues hearing and rotarod performance in the *Srrm4**bv/bv* mouse line."

We agree with the reviewer that the ability of mice to maintain their balance on a rod depends on the functions of multiple organs, not just the balance organs. In this study, we used a fixed (non-rotating) rod to measure how long the mice could maintain their balance on a narrow surface. We revised the Results section to clarify that the mice were placed on a fixed rod.

As suggested by the reviewer, we reworded the quoted sentence to better convey specifically what was measured in the experiments: "Thus, *Tg(DnRest)* rescues hearing and the ability to balance on a fixed rod in the *Srrm4*^{*bv/bv*} mouse line."

- The authors express the data often in this manner: This analysis revealed that 3-4 fold more IHCs survived in *Tg(DnRest);Srrm4**bv/bv* mice than in *Srrm4**bv/bv* littermates (Fig. 1G, Fig. S1F and S1G). I would just state the actual numbers (WT: XX +/- error ; Mutant: YY +/- error, with p-values for stat difference).

We have deleted the fold-change statements from the text because they were redundant with numerical data in Fig. 1. For the same reason, we prefer not to replace the fold-change numbers with mean±SEM and p values. These numbers are shown in Fig. 1 or described in the figure legend. Citing these numbers in the main text would be very lengthy; for example, Fig. 1G shows 9 means and 9 p values for pair-wise comparisons. In addition, the compared data groups would need to be defined in the text for each p value; this would make the Results section difficult to read.

- In Fig. 1F, it looks like the rescue of IHCs in the *Tg(DN)* mice is not 100%. Statistical treatment of data presented in Figure 1G and I would clarify that.

We now include statistical analysis of IHC counts in Fig. 1G. This analysis confirms that *Tg(DnRest)* rescues IHCs in *Srrm4^{bv/bv}* mice. The analysis also reveals that the number of apical-turn IHCs is lower in *Tg(DnRest);Srrm4^{bv/bv}* mice versus WT mice. Although a few IHCs are also missing from the middle and basal turns of the cochlea in *Tg(DnRest);Srrm4^{bv/bv}* mice (Fig. 1F and S1H), statistical analysis does not reveal a significant difference in middle-turn and basal-turn IHC numbers in *Tg(DnRest);Srrm4^{bv/bv}* versus WT mice. Perhaps if we analyzed 5–10 more WT and *Tg(DnRest);Srrm4^{bv/bv}* mice, statistical analysis would confirm a small difference.

- Fig 2A: Here, not only the 8 REST target genes are downregulated in utricular hair cells at E16, but also *Ocm*, which is not known to be a REST target. The easiest explanation would be that the downregulation of 8 + *Ocm* is due to hair cell loss, but the authors exclude that possibility by using E16.5 utricles where there is no obvious hair cell loss. But to really convincingly exclude that HC loss is a factor, it would be good to have one additional (hair cell) control (e.g. *Myo7a*, *Myo6*) where the expression remains unchanged in the *Srrm4^{bv/bv}* mice.

Although it is possible that a few VHCs die by E16.5 in *Srrm4^{bv/bv}* mice, the data in Fig. 2A are unlikely to be significantly affected for two reasons. 1) The qRT-PCR data in Fig. 2A are normalized to the expression level of the HC-specific mRNA *Myo6*. The Materials and Methods section has been revised to describe the normalization of qRT-PCR data in detail. 2) Our new qRT-PCR data reveal that utricular expression levels of *Myo6* (normalized to the housekeeping gene *Heatr3*) differ significantly among WT, *Srrm4^{bv/bv}*, and *Tg(DnRest);Srrm4^{bv/bv}* mice at P60 but not at E16.5. Thus, not all HC-specific transcripts are expressed at abnormally low levels in the utricle of E16.5 *Srrm4^{bv/bv}* mice. These new data have been added to Fig. S2.

- Here is another example where the writing could be improved to aid in the understanding of this rather complex content: The result in Fig 2B is surprising, because the *Tg(DN)* rescues splicing. It would help to explicitly state that this is an unexpected result. Something like: "Thus, *Tg(DnRest)* rescues the alternative splicing of several SRRM4-regulated exons in the *Srrm4^{bv/bv}* mouse. This is surprising, given that REST is not known to regulate pre-mRNA splicing directly. Our data therefore suggest that DnREST rescues alternative splicing in *Srrm4^{bv/bv}* mice through an indirect mechanism. "

We thank the reviewer for the suggested edits. We rephrased this section using some of the wording suggested by the reviewer:

"Thus, *Tg(DnRest)* rescues the alternative splicing of several SRRM4-regulated exons in the utricle of *Srrm4^{bv/bv}* mice. This is surprising because DnREST and REST are not known to regulate pre-mRNA splicing directly. Given that DnREST derepresses the splicing factor-encoding gene *Srrm3* in the utricle of *Srrm4^{bv/bv}* mice (Fig. 2A), the ability of DnREST to rescue alternative splicing could be due to SRRM3."

- In Fig 3B, it is not clear whether the gene trap completely abolishes *Srrm3*, or whether there is some residual expression.

The gene trap does not abolish *Srrm3* expression *completely* because the *Srrm3* mRNA could be detected in *Srrm3^{gt/gt}* samples by qRT-PCR (the difference in *Srrm3* expression between the *Srrm3^{gt/gt}* and WT samples is 50-60-fold). We note that the gene expression levels in Fig. 3B are within the limits of detection of our qRT-PCR assay. When reported data are beyond the limit of detection, we use '>' axis labels (e.g., Fig. 1D) or describe assay limits in the figure legend (e.g., Fig. 1E).

- On page 9, the authors write: We tested whether *Srrm3* expression is necessary for the *Tg(DnRest)*-dependent survival of IHCs and VHCs in *Srrm4^{bv/bv}* mice. *Srrm3* expression was reduced by outcrossing the *Tg(DnRest);Srrm4^{bv/bv}* mouse line to a recently generated line that is heterozygous for a gene trap insertion in *Srrm3* (*Srrm3⁺/gt*). This sounds like the authors are using *Srrm3⁺/gt* mice (although they use the *Srrm3^{gt/gt}*). Just omit the sentence "*Srrm3* expression was reduced by outcrossing the *Tg(DnRest);Srrm4^{bv/bv}* mouse line to a recently generated line that is heterozygous for a gene trap insertion in *Srrm3* (*Srrm3⁺/gt*)." and write that you generated *Tg(DnRest);Srrm3^{gt/gt};Srrm4^{bv/bv}* mice.

We thank the reviewer for the suggested edits. We have incorporated the recommended rewording:

"To reduce *Srrm3* expression in *Tg(DnRest);Srrm4^{bv/bv}* mice, we took advantage of a recently produced gene trap allele of *Srrm3* (*Srrm3^{gt}*) (Nakano et al., 2019). *Tg(DnRest);Srrm3^{gt/gt};Srrm4^{bv/bv}* mice were generated and used for qRT-PCR and histological analyses."

- In Fig.4, the *Tg(DnRest);Srrm3^{gt/gt};Srrm4^{bv/bv}*, there is no survival of OHCs. But as shown in the original Boeda et al and recently in Li et al, 2020 (Myosin-VIIa is expressed in multiple isoforms and essential for tensioning the hair cell mechanotransduction complex), the *Myo7a* promoter does drive some expression in apical OHCs. This means there should be some rescue in the apical OHCs. Have the authors checked that?

Although our immunofluorescence analysis of the *Tg(DnRest)* cochlea did not reveal DnREST expression in OHCs (Fig. 1C), we cannot rule out the possibility that a small amount of DnREST is produced in these cells. Consistent with the reviewer's comment, we did observe morphologically intact OHCs very near the apex of the organ of Corti in DIV9 cultures from some (though not all) *Tg(DnRest);Srrm3^{gt/gt};Srrm4^{bv/bv}* mice. These OHCs were located in the apical ~220-micron segment of the organ of Corti. Given that we counted OHCs along ~2500-micron segments of the organ of Corti, the few non-degenerating OHCs at the very apex did not change the overall numbers much (see in Fig. 4B). Furthermore, in approximately half of the *Tg(DnRest);Srrm3^{gt/gt};Srrm4^{bv/bv}* cultures, OHCs did not remain morphologically intact, even at the very apex of the organ of Corti. We speculate that mouse-to-mouse variation in *Tg(DnRest)* expression levels might account for the sample-to-sample differences in OHC survival at the very apex of the organ of Corti. We further speculate that low expression of *Tg(DnRest)* is insufficient to effectively suppress the activity of endogenous REST. A very sensitive DnREST detection assay will have to be developed to determine whether DnREST is expressed at low levels in OHCs located near the apex of the cochlea.

We think that our data agree well with those reported by Sihan Li and colleagues in their recent study (PMID: 32350269). Specifically, these authors report that "in *Myo7a::Actin-GFP* transgenic mice [...] Actin-GFP signal was predominantly observed in the IHCs. Actin-GFP signal was detected at low levels in the apical OHCs". This paper is now cited in our manuscript.

- First sentence of last paragraph in page 11: We next tested the effect of the SRRM3-SRRM4 double deficiency on expression of the REST target genes that we had found to be downregulated in *Srrm4^{bv/bv}* versus WT utricles (Fig. 2A). Was this supposed to say "in OHCs"?

We have changed this sentence as follows: "We next tested the effect of the SRRM3–SRRM4 double deficiency on the expression of REST target genes in OHCs."

October 9, 2020

RE: Life Science Alliance Manuscript #LSA-2020-00841R

Dr. Botond Banfi
University of Iowa
Anatomy and Cell Biology
431 Newton Road
Iowa City, Iowa 52242

Dear Dr. Banfi,

Thank you for submitting your revised manuscript entitled "Inhibition of a transcriptional repressor rescues hearing in a splicing factor-deficient mouse". We would be happy to publish your paper in Life Science Alliance pending final revisions necessary to meet our formatting guidelines.

Along with the points listed below, please also address to the following:

- please add ORCID ID for corresponding author-you should have received instructions on how to do so
- please provide tables in editable doc or excel format
- please provide source data for all the western blots shown in Figures 2B and 3G

A. FINAL FILES:

-- Summary blurb (enter in submission system): A short text summarizing in a single sentence the study (max. 200 characters including spaces). This text is used in conjunction with the titles of papers, hence should be informative and complementary to the title. It should describe the context and significance of the findings for a general readership; it should be written in the present tense

and refer to the work in the third person. Author names should not be mentioned.

B. MANUSCRIPT ORGANIZATION AND FORMATTING:

Sincerely,

Shachi Bhatt, Ph.D.
Executive Editor
Life Science Alliance
<https://www.life-science-alliance.org/>
Tweet @SciBhatt @LSAjournal

October 12, 2020

RE: Life Science Alliance Manuscript #LSA-2020-00841RR

Dr. Botond Banfi
University of Iowa
Anatomy and Cell Biology
431 Newton Road
Iowa City, Iowa 52242

Dear Dr. Banfi,

Thank you for submitting your Research Article entitled "Inhibition of a transcriptional repressor rescues hearing in a splicing factor-deficient mouse". It is a pleasure to let you know that your manuscript is now accepted for publication in Life Science Alliance. Congratulations on this interesting work.

*****IMPORTANT:** If you will be unreachable at any time, please provide us with the email address of an alternate author. Failure to respond to routine queries may lead to unavoidable delays in publication.*******

DISTRIBUTION OF MATERIALS:

Again, congratulations on a very nice paper. I hope you found the review process to be constructive and are pleased with how the manuscript was handled editorially. We look forward to future exciting submissions from your lab.

Sincerely,

Shachi Bhatt, Ph.D.

Executive Editor

Life Science Alliance

<https://www.lsjournal.org/>
